# Structure and dynamics of the chromatin remodeler ALC1 bound to a PARylated nucleosome

Luka Bacic[1†], Guillaume Gaullier[1†], Anton Sabantsev[1], Laura C Lehmann[1], Klaus Brackmann[1], Despoina Dimakou[1], Mario Halic[2], Graeme Hewitt[3], Simon J Boulton[3], Sebastian Deindl[1]*

[1]Department of Cell and Molecular Biology, Science for Life Laboratory, Uppsala University, Uppsala, Sweden; [2]Department of Structural Biology, St Jude Children's Research Hospital, Memphis, United States; [3]The Francis Crick Institute, London, United Kingdom

**Abstract** The chromatin remodeler ALC1 is recruited to and activated by DNA damage-induced poly(ADP-ribose) (PAR) chains deposited by PARP1/PARP2/HPF1 upon detection of DNA lesions. ALC1 has emerged as a candidate drug target for cancer therapy as its loss confers synthetic lethality in homologous recombination-deficient cells. However, structure-based drug design and molecular analysis of ALC1 have been hindered by the requirement for PARylation and the highly heterogeneous nature of this post-translational modification. Here, we reconstituted an ALC1 and PARylated nucleosome complex modified in vitro using PARP2 and HPF1. This complex was amenable to cryo-EM structure determination without cross-linking, which enabled visualization of several intermediate states of ALC1 from the recognition of the PARylated nucleosome to the tight binding and activation of the remodeler. Functional biochemical assays with PARylated nucleosomes highlight the importance of nucleosomal epitopes for productive remodeling and suggest that ALC1 preferentially slides nucleosomes away from DNA breaks.

*For correspondence: sebastian.deindl@icm.uu.se

[†]These authors contributed equally to this work

## Introduction

DNA double-strand breaks (DSBs) are among the most cytotoxic DNA lesions, and a common strategy in cancer therapy is to overwhelm the repair capacity of cancer cells with excess DSBs using radiation or cytotoxic chemotherapies. DSBs can only be repaired successfully if they are first recognized, and their recognition elicits a DNA damage signaling cascade. One of the earliest components of the DNA damage response (DDR) able to recognize DSBs are the ADP-ribosyltransferases PARP1 and PARP2 (*Krishnakumar and Kraus, 2010*; *Liu et al., 2017b*; *Lüscher et al., 2021*). Targeted inhibition of the DNA damage response in cancer cells is often used alone or in combination to augment the cytotoxic effect of DSBs, as exemplified by clinical PARP inhibitors.

Chromatin remodeling by ALC1 (Amplified in Liver Cancer 1) plays an important role during the early stages of the DDR elicited by poly-ADP-ribosylation (PARylation) at DNA lesions (*Sellou et al., 2016*). ALC1 uses its macro domain to bind to PAR (*Ahel et al., 2009*; *Gottschalk et al., 2009*) produced by PARP1 and PARP2. In the absence of DNA damage, the macro domain of ALC1 is abutted against its ATPase, which stabilizes an inactive conformation (*Lehmann et al., 2017*; *Singh et al., 2017*). Structural, biochemical, and in vivo analyses suggested that the macro domain interacts predominantly with the C-terminal ATPase lobe (*Lehmann et al., 2017*). PAR binding to the macro domain relieves this auto-inhibition upon recruitment to DNA damage (*Lehmann et al., 2017*; *Singh et al., 2017*). Full activation of ALC1 requires its linker to insert an Arg anchor motif into the 'acidic

patch' (AP) composed of negatively charged residues from histones H2A and H2B on both faces of the nucleosome (*Lehmann et al., 2020*). A PARylation response leading to efficient repair requires HPF1 (Histone PARylation Factor 1), since cells lacking HPF1 display much-reduced survival after DNA damage (*Bonfiglio et al., 2017*). As a crucial step in the DNA damage response, activation of ALC1 therefore most likely requires nucleosomal histone PARylation in the presence of HPF1.

Recent studies have defined ALC1 as an attractive target for therapeutic intervention strategies in cancer as its inactivation sensitizes to clinical PARP inhibitors and confers synthetic lethality in homologous recombination-deficient cancer cells (*Abbott et al., 2020*; *Blessing et al., 2020*; *Hewitt et al., 2021*; *Juhász et al., 2020*; *Verma et al., 2021*). However, despite thorough biochemical and biophysical scrutiny of its regulation and interaction with nucleosomes (*Ahel et al., 2009*; *Gottschalk et al., 2012*; *Gottschalk et al., 2009*; *Lehmann et al., 2020*; *Lehmann et al., 2017*; *Singh et al., 2017*), ALC1 has so far resisted structure determination.

Most high-resolution cryo-EM structures of nucleosome-bound chromatin remodelers available to date were obtained with cross-linked complexes (*Baker et al., 2021*; *Farnung et al., 2020*; *Farnung et al., 2017*; *Han et al., 2020*; *He et al., 2020*; *Meijing et al., 2019*; *Liu et al., 2017a*; *Patel et al., 2019*; *Wagner et al., 2020*; *Yan et al., 2019*; *Ye et al., 2019*). Nonetheless, several structures of non-cross-linked nucleosome-remodeler complexes have attained resolutions of around 4 Å, sufficient to unambiguously assign secondary structure elements and to provide important mechanistic insights. Of these, three were multi-subunit remodeling complexes (*Ayala et al., 2018*; *Eustermann et al., 2018*; *Willhoft et al., 2018*) that may be overall more stable due to multiple interactions between catalytic and scaffolding subunits and the nucleosome, and the remaining three were single-subunit remodelers (*Armache et al., 2019*; *Chittori et al., 2019*; *Sundaramoorthy et al., 2018*). PARylation is intrinsically heterogeneous, and ALC1 physically interacts with at least PAR chains and the nucleosome acidic patch in addition to the nucleosomal DNA. The recognition of a PARylated nucleosome by ALC1 therefore likely involves stochastically probing these epitopes via a variety of structural states until settling into the active conformation competent for remodeling. Consequently, cross-linking the complex may obscure key information on the recognition process, by turning continuous conformational flexibility into a set of discrete states. Modeling continuous conformational heterogeneity from cryo-EM data is still an open research question, but recent advances in software development suggest that it is now a tractable problem, and that one can circumvent cross-linking (*Punjani and Fleet, 2021*; *Zhong et al., 2021*).

Here, we show that PARylated nucleosomes can be efficiently produced in vitro with PARP1 or PARP2 and HPF1 and that such nucleosomes are the optimal substrate for ALC1. We reconstituted a complex between ALC1 and a PARylated nucleosome. In the absence of cross-linking, this complex yielded cryo-EM data of sufficient quality not only to permit structure determination of the active state of the nucleosome-bound remodeler, but also to visualize multiple conformational states visited during substrate recognition. Examination of these states identified physical interactions between ALC1 and several nucleosomal epitopes, which our functional assays confirm to be critical for remodeling.

## Results

### The optimal substrate of ALC1 is a PARylated nucleosome

For structure determination by cryo-EM, we set out to form a stable complex between ALC1 and a nucleosome. At concentrations suitable for cryo-EM sample preparation, the isolated ATPase motor of ALC1 (ALC1[cat]) did not yield any detectable binding to nucleosomes in a gel shift assay (*Figure 1A*, lane 2). While an essentially full-length protein (ALC1[fl]) did bind to these nucleosomes (*Figure 1A*, lane 4), this binding predominantly originated from non-specific interactions between its macro domain (ALC1[macro]) and DNA (*Figure 1A*, lane 3), in agreement with our previous results (*Lehmann et al., 2017*). In order to explore the impact of nucleosome PARylation on complex formation with ALC1, we repeated these experiments after PARylating the nucleosomes with PARP2 and HPF1 (*Bonfiglio et al., 2017*). ALC1[cat] was still unable to bind the PARylated nucleosomes (*Figure 1A*, lanes 5 and 6), while ALC1[macro] interacted strongly (*Figure 1A*, lane 7), consistent with the previously reported $K_D$ of ~11 nM for the interaction between ALC1[macro] and a tri-(ADP-ribose) model of PAR chains (*Singh et al., 2017*). The binding of ALC1[fl] to PARylated nucleosomes resulted in a more defined pattern

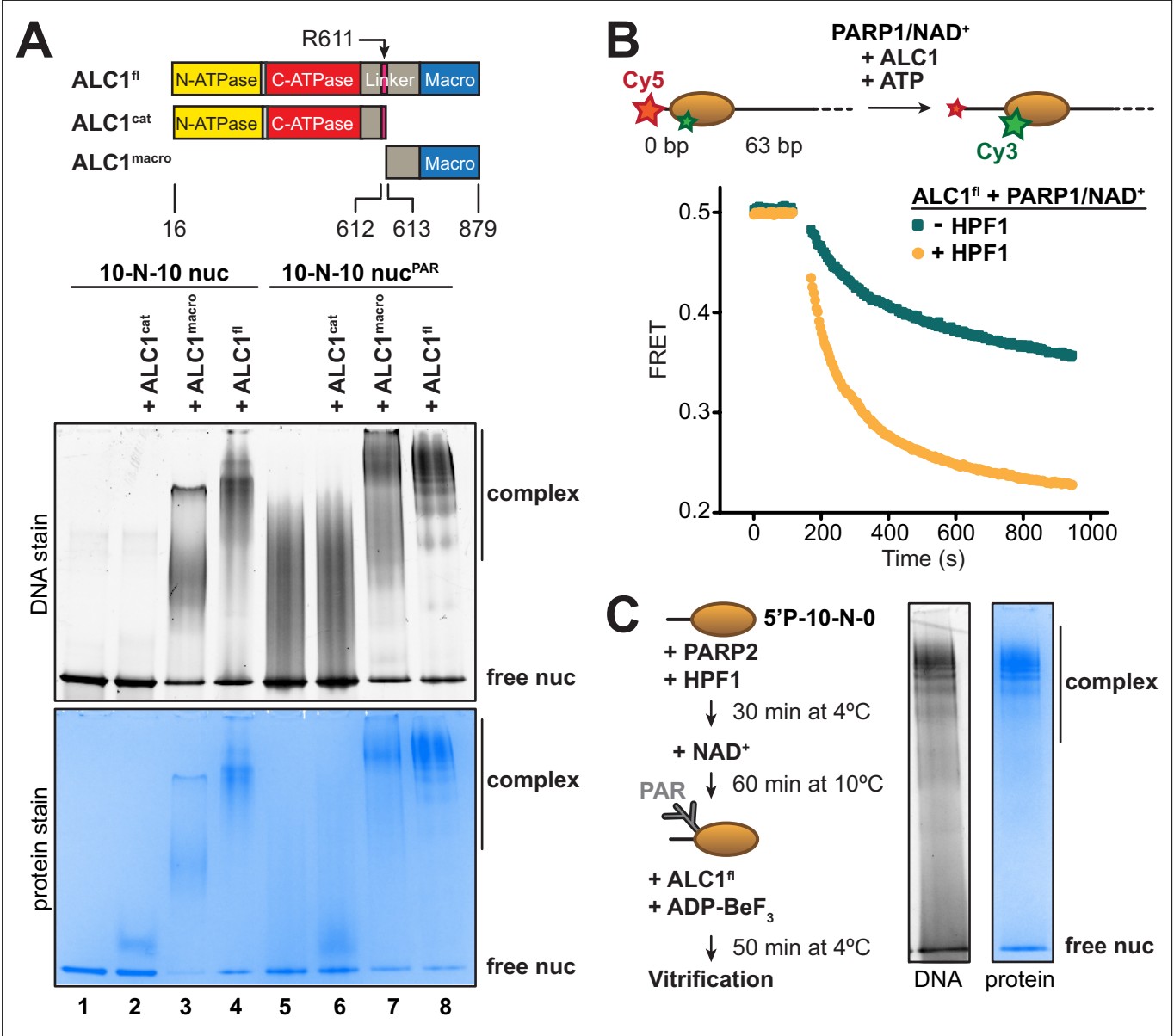

**Figure 1.** Identification of an ALC1-PARylated nucleosome complex suitable for cryo-EM. (**A**) Top: schematic of ALC1 constructs used in EMSA experiments. The arginine anchor residue R611, previously shown to interact with the nucleosome acidic patch, is indicated. Bottom: EMSA analysis of the complexes formed by different constructs of ALC1 and a 5'P-10-N-10 nucleosome in absence or presence of PAR chains deposited by PARP2 and HPF1. (**B**) Nucleosome sliding assay of 10 nM nucleosomes by 31.3 nM ALC1$^{fl}$, performed after PARylation by 80 nM of PARP1 with 25 μM NAD$^+$ in the absence (teal) and presence (orange) of 20 nM of HPF1. (**C**) Preparation and native PAGE analysis of the complex between ALC1$^{fl}$ and a PARylated nucleosome for cryo-EM (see Materials and methods).

The online version of this article includes the following source data and figure supplement(s) for figure 1:

**Source data 1.** Tiff files of raw gel images for *Figure 1A,C*.

**Figure supplement 1.** Nucleosome sliding assay replicated as shown in *Figure 1B*.

**Figure supplement 2.** Nucleosome sliding assays with 5 min (orange) or 30 min (brown) incubation time of the PARylation reaction prior to the initiation of the sliding reaction.

**Figure supplement 3.** SDS–PAGE and Western blot analysis of different PARylation conditions of nucleosomes by PARP1/HPF1 or PARP2/HPF1.

**Figure supplement 4.** Screening micrographs of complexes prepared with ALC1 and nucleosomes either unmodified (left), PARylated by PARP1 and HPF1 (middle), or PARylated by PARP2 and HPF1 (right).

of shifted bands (*Figure 1A*, lane 8 compared to lane 4), suggesting a complex with more restricted conformations.

To determine whether the complexes with PARylated or unmodified nucleosomes (*Figure 1A*, lanes 4 and 8) were functionally different, we adapted a fluorescence resonance energy transfer (FRET)-based nucleosome sliding assay (*Yang et al., 2006*). In a previous study, we reported the addition of pre-PARylated PARP1 (auto-modified upon activation by free DNA) to activate sliding by ALC1 (*Lehmann et al., 2017*). Here we instead used the fluorescently labeled and end-positioned nucleosomes to directly activate PARP1, to generate PAR chains in a first reaction (referred to as the PARylation reaction). In a second reaction, we added ALC1 and ATP to initiate nucleosome sliding (referred to as the nucleosome sliding reaction). We first tested the effect of including or omitting HPF1 in the PARylation reaction, leading to PAR chains on both histones and PARP1 or on PARP1 alone, respectively (*Bonfiglio et al., 2017*). We verified by Western blotting that the inclusion of HPF1 in the PARylation reaction did in fact install PAR chains on histones (*Figure 1—figure supplement 1*). We also tested the effect of different incubation times for the PARylation reaction, which showed that a shorter incubation time produced very similar nucleosome sliding rates when compared to a longer incubation (*Figure 1—figure supplement 2*). Nucleosome sliding by ALC1 was ~2.4 -fold faster when PAR chains were attached to histones, rather than exclusively to PARP1 (*Figure 1B*), indicating that the PARylated nucleosome is the proper substrate of ALC1. This observation also corroborates the paradigm that in-trans ADP-ribosylation of histones, and not only auto-modification of PARP1 and PARP2, is critical to elicit a DNA damage response leading to successful repair and cellular viability (*Bonfiglio et al., 2017*).

Next, we designed an ALC1-PARylated nucleosome complex amenable to cryo-EM analyses based on the following considerations. The structure of PARP2 and HPF1 bound to nucleosomes indicates that, of the two H3 tails in a nucleosome, the one on the side of the DNA end bound by PARP2 is closest to its active site (*Bilokapic et al., 2020*). Target residues for ADP-ribosylation (Ser in KS motifs) in this proximal H3 tail should therefore be favored over the equivalent residues in the distal H3 tail. We reasoned that controlling which DNA end PARP2 binds to may yield preferred PARylation of one of the two H3 tails, which may in turn cause ALC1 to preferentially bind to that side of the nucleosome. Conveniently, PARP2 requires a terminal 5'-phosphate group for binding and activation (*Langelier et al., 2014*; *Obaji et al., 2018*). We therefore prepared a nucleosome with a 5'-phosphate group at the end of a 10 bp linker DNA on one side, and with a free 5'-hydroxyl at the end of a 0 bp linker DNA on the other side (termed 5'P-10-N-0 nucleosome). This design should strongly favor PARP2 binding to the 10 bp linker, since the 0 bp linker is devoid of a terminal 5'-phosphate. We assessed PARylation conditions to determine the concentrations of PARP2, HPF1 and NAD$^+$ that enable PARylation of histones but also limit the auto-modification of PARP2 (*Figure 1—figure supplement 3*). All our attempts were aimed at limiting PAR chain length because it is known that as few as three ADP-ribose units are sufficient for tight binding and full activation of ALC1 (*Singh et al., 2017*). While systematically assessing conditions with PARP1, we realized that it indiscriminately produced PAR chains sufficiently long and heterogeneous to compromise cryo-EM image analysis (*Figure 1—figure supplement 3* and *Figure 1—figure supplement 4*). We therefore turned to PARP2, as it produced noticeably shorter PAR chains (*Figure 1—figure supplement 3*) and yielded higher-quality particles (*Figure 1—figure supplement 4*). We noticed that high concentrations of HPF1 tend to limit the elongation of PAR chains, a phenomenon also recently reported, investigated in detail, and exploited by others (*Langelier et al., 2021*; *Mohapatra et al., 2021*). Screening micrographs and exploratory 2D and 3D classifications from small screening datasets confirmed that the ALC1-nucleosome complex most amenable to the collection of a large dataset was the one formed with ALC1$^{fl}$ and a 5'P-10-N-0 nucleosome PARylated by PARP2 and HPF1 (*Figure 1C*, *Figure 1—figure supplement 4*).

## Cryo-EM structure of the complex between a PARylated nucleosome and ALC1 in its active state

For structure determination, we collected cryo-EM data of a non-cross-linked complex between ALC1$^{fl}$ and a 5'P-10-N-0 nucleosome PARylated by PARP2 and HPF1. Despite a high level of conformational heterogeneity in our dataset (*Figure 2—figure supplement 1*), we could isolate a set of 5487 particles that yielded a map of the ATPase domain of ALC1 bound to the nucleosome with a global resolution of 4.8 Å (*Figure 2A*, *Figure 2—figure supplement 1* and *Table 1*). In the absence

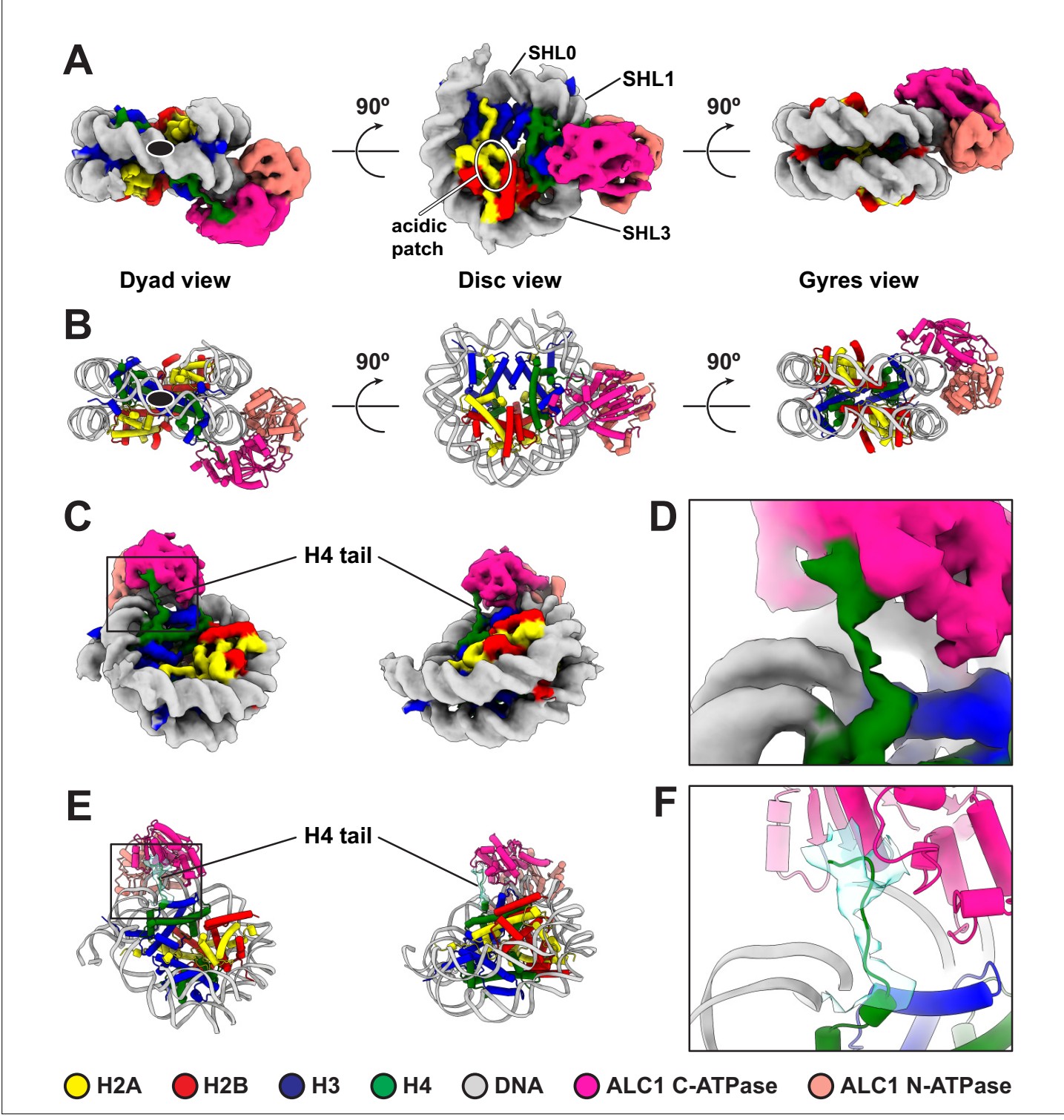

H2A   H2B   H3   H4   DNA   ALC1 C-ATPase   ALC1 N-ATPase

**Figure 2.** Cryo-EM structure of the complex between a PARylated nucleosome and ALC1 in its active state. (**A**) Cryo-EM map of the complex between a PARylated nucleosome and ALC1 in the state tightly bound to SHL2 and the H4 tail. The map is shown at a contour level of 0.15 and colored by chain assignment (H3 in blue, H4 in green, H2A in yellow, H2B in red, DNA in gray, and ALC1 in pink). The black ellipse in the dyad view indicates the nucleosome dyad axis. Superhelical locations 0 (dyad), 1, and 3 and the location of the acidic patch are indicated. (**B**) Atomic model of the complex between a PARylated nucleosome and ALC1 in the state tightly bound to SHL2 and the H4 tail. Same views and same color code as in (**A**). The black ellipse in the dyad view indicates the nucleosome dyad axis. (**C**) Same cryo-EM map as in (**A**), displayed at the same contour level, but in two different orientations, showing the H4 tail interacting with the C-terminal ATPase domain of ALC1. Same color code as in (**A**). (**D**) Close-up view of the boxed

*Figure 2 continued on next page*

*Figure 2 continued*

region in (**C**). (**E**) Atomic model shown in the same two orientations as in (**C**). In addition to the same color code as in (**A**), the cryo-EM density of the H4 tail is shown as a turquoise transparent surface, at the same contour level as in (**A**), (**C**), and (**D**). (**F**) Close-up view of the boxed region in (**E**).

The online version of this article includes the following figure supplement(s) for figure 2:

**Figure supplement 1.** Overview of cryo-EM data processing and analysis.

**Figure supplement 2.** Superimposition of previous structures of remodeler-nucleosome complexes to our structure of the ALC1-nucleosome complex.

**Figure supplement 3.** Close-up view of the structure of the complex between a PARylated nucleosome and ALC1 in the tightly bound state, showing the H4 tail interacting with the C-terminal ATPase domain of ALC1.

**Table 1.** Cryo-EM data collection and refinement statistics.

Data collection and processing

| | | |
|---|---|---|
| Acceleration voltage (kV) | 300 | |
| Spherical aberration (mm) | 2.7 | |
| Amplitude contrast (fraction) | 0.1 | |
| Image pixel size (Å/pixel) | 0.84 | |
| Electron exposure per frame (e⁻/Å²) | 1.125 | |
| Number of movie frames | 40 | |
| Total electron exposure (e⁻/Å²) | 45 | |
| Nominal defocus range (μm) | –1.0 to –2.5 | |
| Number of movies collected | 26,747 | |
| Number of picked particles | 3,321,590 | |
| Particles used for reconstruction | 5487 (single map) | 43 698 (cryoDRGN analysis) |
| Map symmetry imposed | C1 | |
| Map sharpening B-factor (Å²) | 128.2 | |
| Global resolution at 0.143 FSC (Å) | 4.8 | NA |
| Local resolution range (Å) | 4–20 | |

**Model building and refinement**

| | | |
|---|---|---|
| Initial models used (PDB codes) | Nucleosome (6RYR) Template for homology modeling of ALC1 ATPase (6JYL) | |
| Number of atoms (hydrogens) | 29,161 (13 383) | |
| Number of protein residues | 1 214 | |
| Number of DNA residues | 298 | |
| Bond length RMSD (Å) | 0.004 | |
| Bond angles RMSD (°) | 0.760 | |
| MolProbity score | 1.74 | NA |
| Clash score | 6.69 | |
| Ramachandran outliers/allowed/favored (%) | 0.25/3.68/96.07 | |
| Rotamer outliers (%) | 1.36 | |
| Cβ outliers (%) | 0.00 | |
| CαBLAM outliers (%) | 2.72 | |
| Model to map CC (mask/box/peaks/volume) | 0.77/0.84/0.71/0.77 | |

of cross-linking, only a small number of particles adopted this stable conformation, which limits the attainable resolution. Nevertheless, this resolution is sufficient for the map to display clear secondary structure elements, allowing model building (*Figure 2B*). The map reveals an ATPase motor tightly bound to the nucleosomal DNA at superhelical location (SHL) 2 (*Figure 2A,B*), where the second major groove faces the histone octamer when numbering DNA turns starting at the nucleosome dyad (reviewed in *McGinty and Tan, 2015*; *Zhou et al., 2019*). Chd1, ISWI, and SWI/SNF-type remodelers all translocate on nucleosomal DNA at this SHL2 site (*Deindl et al., 2013*; *McKnight et al., 2011*; *Saha et al., 2005*; *Schwanbeck et al., 2004*; *Zofall et al., 2006*). The map also shows a clear inter-action of the C-terminal ATPase lobe with the N-terminal tail of histone H4 (*Figure 2C–F*). Both of these features are conserved across almost all chromatin remodelers structurally characterized to date (*Armache et al., 2019*; *Chittori et al., 2019*; *Farnung et al., 2020*; *Farnung et al., 2017*; *Meijing et al., 2019*; *Liu et al., 2017a*; *Sundaramoorthy et al., 2018*; *Wagner et al., 2020*; *Yan et al., 2019*; *Ye et al., 2019*).

To further examine the conformation of the ALC1 ATPase motor, we compared our model to published structures of nucleosome-bound remodelers from each family of single-subunit remod-elers with or without a nucleotide ligand (ADP or ADP-BeF$_3$): Snf2, Isw1, and CHD4 (*Farnung et al., 2020*; *Meijing et al., 2019*; *Yan et al., 2019*). We superimposed each of these structures (PDB entries 5Z3L, 5Z3U, 5Z3O, 6JYL, 6IRO, and 6RYR) to our model, based on the histone H3 coordinates, which revealed differences in ATPase motor conformations (*Figure 2—figure supplement 2*). The ATPase motor of ALC1 aligns optimally with the ADP-BeF$_3$-bound conformations of Snf2 and Isw1, compared to their conformations in the presence of ADP or in the absence of any ligand (for Snf2), consistent with our use of this ligand to induce the active conformation of the motor. The ATPase motor of ALC1 is otherwise most similar to that of its closest homologue CHD4.

Notably, our map does not show density for either the macro domain of ALC1 or PAR chains on the nucleosome, most likely due to their extreme conformational flexibility in the active state (*Figure 2A*). However, the average map from a consensus 3D refinement of all 43,698 particles displays clear density for an interaction between ALC1 and the nucleosome acidic patch (*Figure 3A*, *Figure 2—figure supplement 1*), confirming at the structural level and in the context of ALC1$^{fl}$ our previous finding that the acidic patch is important for remodeling by ALC1 (*Lehmann et al., 2020*). Moreover, this map also features density that we assign to the macro domain (*Figure 3A*, *Figure 2—figure supplement 1*) based on the following rationale: this density makes contact with the ATPase motor, in agreement with the previously reported ATPase motor-macro domain interaction (*Lehmann et al., 2017*; *Singh et al., 2017*). Moreover, the density resides on the side of ALC1 directed towards SHL1 where the N-terminal tail of H3 emerges from the two DNA gyres of the nucleosome. The macro domain would thus be positioned at a location compatible with binding to PAR chains at the main PARylation site of the nucleosome (*Bonfiglio et al., 2017*). Comparing the map of the active state (*Figure 2A*) with the average map from the larger particle set (*Figure 3A*, *Figure 2—figure supplement 1*) also hints at structural information on additional functional states of the complex.

## Analysis of heterogeneity in the cryo-EM data reveals additional functional states of ALC1

Because both 2D classification and 3D variability analysis uncovered continuous conformational heterogeneity in the dataset (*Figure 2—figure supplement 1*), we analyzed it with cryoDRGN (*Zhong et al., 2021*). The distribution of particles in the latent space is smooth, with no discrete clusters, confirming the heterogeneity to be of a purely continuous conformational nature (*Figure 3A*). Repli-cate training runs performed with independent initializations and different network sizes produced comparable results (*Figure 3—figure supplement 1*, *Table 2*). Principal component analysis of the Z values assigned to all particles revealed that the two opposite states along the first principal compo-nent of variability correspond to a state with ALC1 tightly bound to SHL2 and the H4 tail (same state described in *Figure 2*) and a state in which the ATPase domain is only loosely bound to the nucleo-somal DNA (*Videos 1–3*). We used the graph traversal algorithm implemented in cryoDRGN (*Zhong et al., 2021*) with intermediate steps chosen to traverse the distribution along the first two prin-cipal components, which run approximately parallel to the two UMAP axes in this case (*Figure 3A*). Because cryo-EM data do not contain temporal information per se, this approach cannot inform on kinetics and temporal order. However, because each step of the resulting graph traversal trajectory is

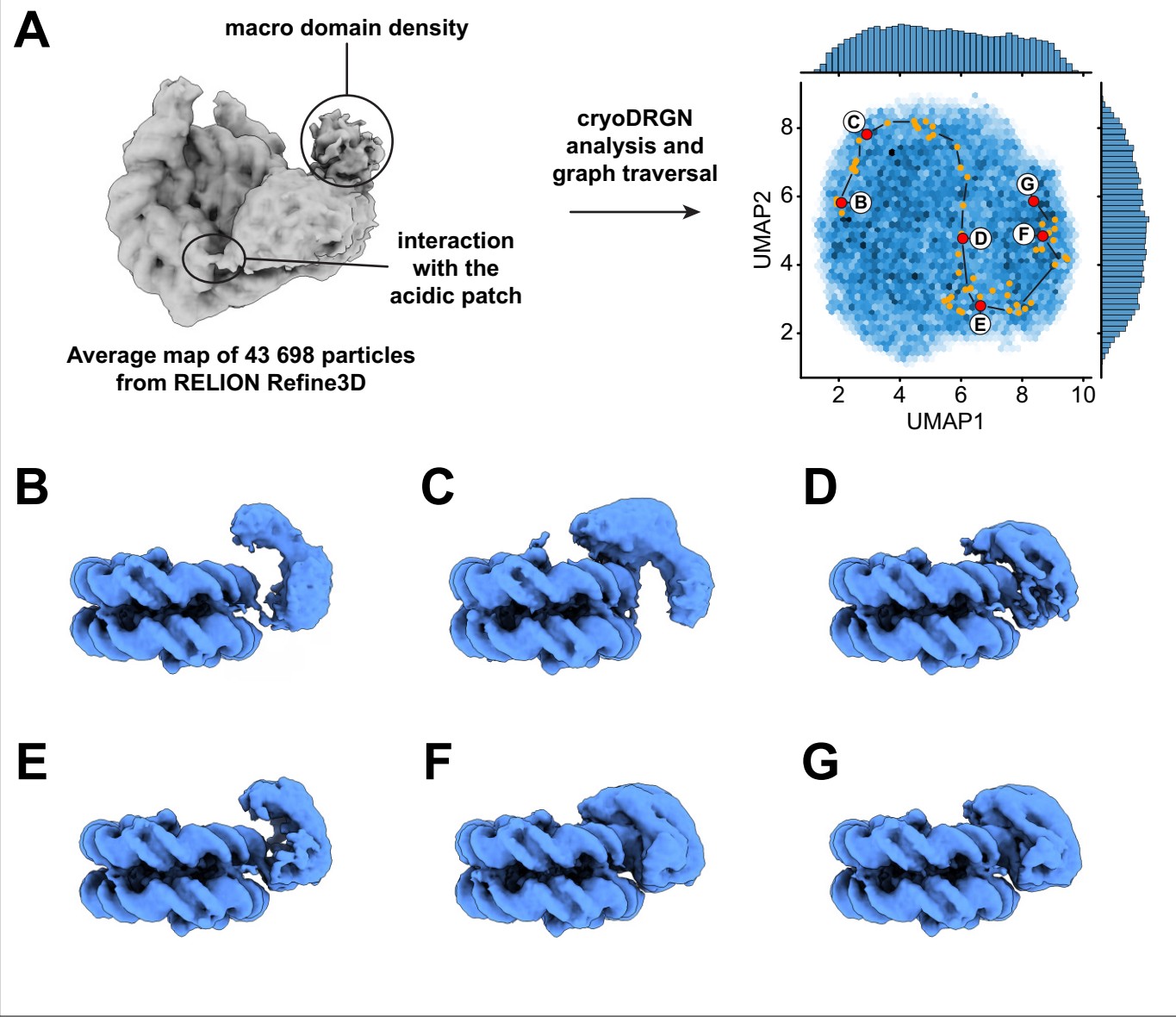

**Figure 3.** Dynamics of the ATPase domain of ALC1. (**A**) Left: consensus map from 3D refinement in RELION of 43,698 particles, displayed at a contour level of 0.015 and in disc view. Regions of the map assigned to an interaction with the nucleosome acidic patch and to the macro domain are labeled. Right: UMAP visualization of the latent variable distribution of these same 43,698 particle images after training of an 8-dimensional latent variable model with cryoDRGN. Dots indicate all latent space coordinates that are part of the calculated graph traversal, labeled red dots indicate coordinates of maps shown in panels (**B**) – (**G**). Black lines are a visual guide and do not represent actual connectivity of the graph traversal. (**B** – **G**) Maps sampled at representative steps along the graph traversal shown in (**A**). All maps are displayed at a contour level of 0.025 and in gyres view.

The online version of this article includes the following figure supplement(s) for figure 3:

**Figure supplement 1.** UMAP visualizations of the latent variable distributions from three replicate training runs of cryoDRGN with independent initializations, and with the indicated image size, pixel size, network architecture, and dimensionality of the latent variable.

**Figure supplement 2.** Atomic model of auto-inhibited ALC1 (PDB 7EPU) rigid-body fitted into one of the maps from cryoDRGN with an open conformation of the ATPase motor.

**Figure supplement 3.** Distribution of Z values from a one-dimensional latent variable training run of cryoDRGN.

supported by observed data, graph traversal enables visualization of plausible conformations visited between these two states (*Zhong et al., 2021*).

The ATPase domain is highly dynamic along the entire trajectory (*Figure 3*, *Video 4*). Strikingly, we detected states in which the two ATPase lobes are splayed apart in a wide-open conformation

**Table 2.** Settings of cryoDRGN training runs.

| Number of particles | Image size (pixels) | Pixel size (Å/pixel) | Image size (Å) | Dimension of Z | Network architecture | Number of epochs | Goal |
|---|---|---|---|---|---|---|---|
| 43,698 | 128 | 3.36 | 430 | 1 | 256 × 3 | 50 | Detection of junk |
| 43,698 | 128 | 3.36 | 430 | 8 | 256 × 3 | 50 | Evaluation (rep. 1) |
| 43,698 | 128 | 3.36 | 430 | 8 | 256 × 3 | 50 | Evaluation (rep. 2) |
| 43,698 | 128 | 3.36 | 430 | 8 | 256 × 3 | 50 | Evaluation (rep. 3) |
| 43,698 | 128 | 1.68 | 215 | 8 | 256 × 3 | 50 | Higher res. 1 |
| 43,698 | 128 | 1.68 | 215 | 8 | 512 × 3 | 50 | Higher res. 2 |

(*Figure 3B,C*), similar to the auto-inhibited ATPase motor conformation inferred from SAXS and cross-linking experiments performed in the absence of a nucleosome (*Lehmann et al., 2017*). In intermediate states, the ATPase domain adopts a closed conformation that resembles the active state conformation but is not as tightly bound to the nucleosomal DNA (*Figure 3D,E*). Finally, cryoDRGN identified states with a closed and tightly bound ATPase motor (*Figure 3F,G*) identical to the active state identified independently using cryoSPARC 3D variability analysis (*Punjani and Fleet, 2021*; *Figure 2A*, *Figure 2—figure supplement 1*). While the ATPase domain adopts its closed conformation and becomes tightly bound to SHL2, it also samples the nucleosomal DNA around SHL2, approximately from SHL1 to SHL3 (*Figure 4*, *Videos 5 and 6*).

The states with an open, loosely bound ATPase domain also feature an interaction with the nucleosome acidic patch (*Figure 4* and *Video 5*). This observation is consistent with our earlier discovery of an interaction between the linker region of ALC1 and the acidic patch, and the role of this interaction in regulating ALC1 (*Lehmann et al., 2020*). Moreover, the density attributed to the macro domain is visible in all these states. Both the acidic patch interaction and the macro domain densities progressively fade away as the ATPase domain transitions to the tightly bound state (*Figure 4D–G*, *Video 5*), indicating that the macro domain and linker occupy less constrained positions when the ATPase domain is in its active conformation. This is in agreement with the model proposed previously, whereby activation of the ATPase requires the disruption of the auto-inhibitory interaction with the macro domain (*Lehmann et al., 2017*; *Singh et al., 2017*). We could further detect states with density connecting the nucleosome and macro domain, at a location consistent with the PARylated H3 tail (*Figure 4C*; also detected in an independent training run, *Figure 4—figure supplement 1*). Finally, the

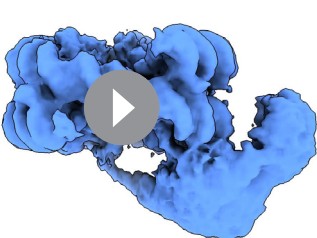

**Video 1.** Traversal of the first principal component of variability in the Z values assigned by cryoDRGN, with maps shown in dyad view.

https://elifesciences.org/articles/71420/figures#video1

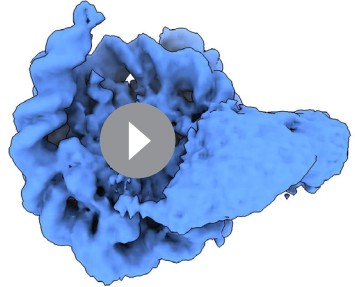

**Video 2.** Traversal of the first principal component of variability in the Z values assigned by cryoDRGN, with maps shown in disc view.

https://elifesciences.org/articles/71420/figures#video2

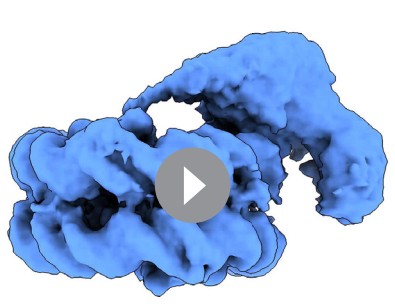

**Video 3.** Traversal of the first principal component of variability in the Z values assigned by cryoDRGN, with maps shown in gyres view.
https://elifesciences.org/articles/71420/figures#video3

interaction between the C-terminal ATPase lobe of ALC1 and the N-terminal tail of histone H4 is detected in most states, including some in which the ATPase domain is otherwise loosely bound to the DNA (*Video 6*). This suggests that this interaction plays a role in recognition, in addition to stimulating the catalytic activity of the ATPase domain as previously shown for other remodelers (*Clapier and Cairns, 2012*; *Hwang et al., 2014*; *Ludwigsen et al., 2017*; *Markert et al., 2021*; *Racki et al., 2014*).

Importantly, graph traversal trajectories calculated with different intermediate steps used as anchors reproduced transitions between very similar states. The trajectory presented here therefore likely visualizes the recognition of a PARylated nucleosome by ALC1 leading to its activation (*Figures 3 and 4*, *Videos 4–6*).

## Nucleosome remodeling by ALC1 depends on its interactions with the acidic patch and the H4 tail

Our structural analysis points to interactions between ALC1 and the PARylated nucleosome as potentially important for productive remodeling; namely interactions between the macro domain and the PAR chains, between the C-ATPase lobe and the H4 tail, and between the regulatory linker of ALC1 and the nucleosome acidic patch. Previous experiments using PARylated PARP1 or tri-(ADP-ribose) had suggested binding of the macro domain to PAR chains to be required for ATPase activation (*Lehmann et al., 2017*; *Singh et al., 2017*). When stimulated by an unmodified nucleosome, even a constitutively active mutant of ALC1, in which the macro domain does not fully inhibit the ATPase (ALC1$^{fl}$ R860W), exhibited reduced ATP hydrolysis rates when compared to a prototypical remodeler like Chd1 (*Hauk et al., 2010*; *Lehmann et al., 2020*; *Sundaramoorthy et al., 2017*). Because the interaction between the macro domain and PAR chains is critical for both binding and activation, we anticipated that disrupting this interface would only result in a fully inactive enzyme. Therefore, we decided to measure nucleosome sliding with PARylated nucleosomes and an unaltered macro domain, while disrupting either the interaction with the H4 tail or that with the acidic patch.

We conducted nucleosome sliding experiments at varying concentrations of ALC1, with or without HPF1 in the PARylation reaction prior to measuring the sliding rates (*Figure 5A*). When PAR chains were attached to PARP1 but not to the nucleosome (in the absence of HPF1), the sliding reaction followed Michaelis–Menten kinetics with a $K_M$ of 30.7 nM and a $V_{max}$ of 0.11 (a.u.) (*Figure 5B*). Unexpectedly, this was not the case when PAR chains were attached to the nucleosome (*Figure 5B*). One likely explanation is that the 63 -N-0 end-positioned nucleosomes were preferentially PARylated on the H3 tail proximal to the short DNA end, since the PARP1 molecule bound at the end of the 63 bp linker DNA is too far away to reach the other H3 tail (approximately 214.2 Å away, equivalent to twice the diameter of a nucleosome and therefore too long to be spanned by even a fully extended H3 tail). Asymmetric nucleosome PARylation would result in two binding sites for ALC1 with different affinities: the PARylated side at SHL+2 with high affinity for ALC1, and the unmodified side at SHL-2 with lower affinity for ALC1. At increasing concentrations of ALC1, the PARylated side would become occupied first and ALC1 would consequently slide the histone octamer away from the Cy5-labeled end. At sufficiently high concentrations of ALC1, a second molecule could engage the weaker second binding site and catalyze sliding in the opposite direction, thereby

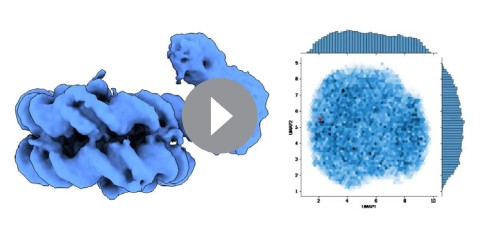

**Video 4.** Dynamics of the ATPase lobes of ALC1.
https://elifesciences.org/articles/71420/figures#video4

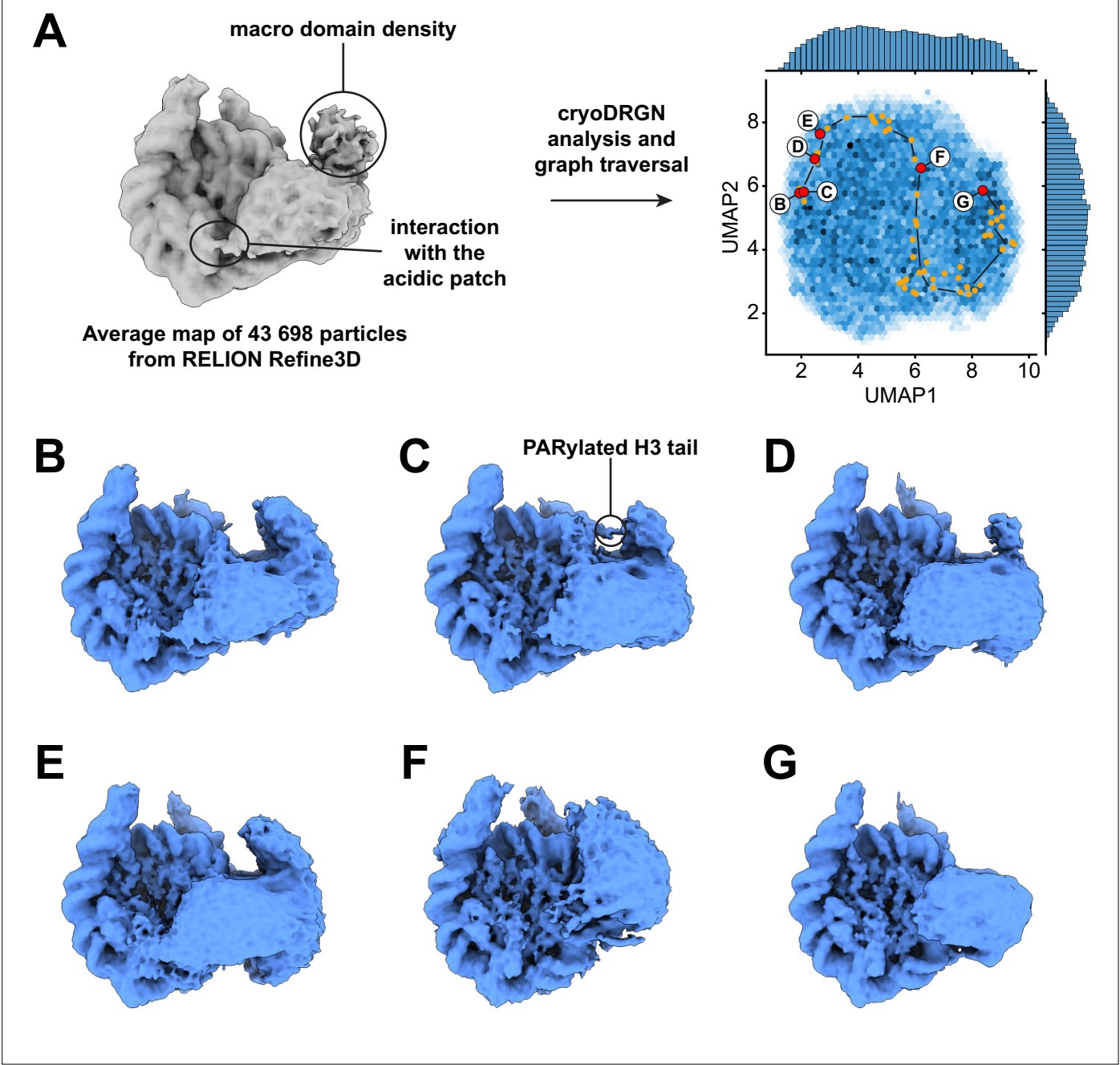

**Figure 4.** Dynamics of the acidic patch interacting region and macro domain of ALC1. (**A**) Left: consensus map from 3D refinement in RELION of 43,698 particles, displayed at a contour level of 0.015 and in disc view. Regions of the map assigned to an interaction with the nucleosome acidic patch and to the macro domain are labeled (same panel as in *Figure 3A*). Right: same plot as in *Figure 3A* but sampling different steps along the same graph traversal. Dots indicate all latent space coordinates that are part of the calculated graph traversal, labeled red dots indicate coordinates of maps shown in panels (**B**) – (**G**). Black lines are a visual guide and do not represent actual connectivity of the graph traversal. (**B**) – (**G**) Maps sampled along the graph traversal shown in (**A**). All maps are displayed at a contour level of 0.013 and in disc view.

The online version of this article includes the following figure supplement(s) for figure 4:

**Figure supplement 1.** UMAP visualization of the latent variable distribution from a replicate training run of cryoDRGN with independent initialization and with the indicated image size, pixel size, network architecture, and dimensionality of the latent variable.

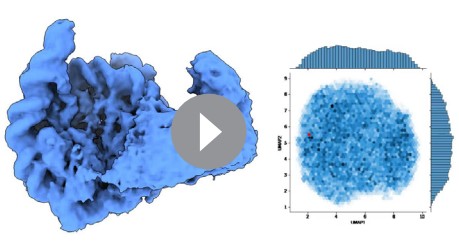

**Video 5.** Dynamics of the acidic patch interacting region and macro domain of ALC1.
https://elifesciences.org/articles/71420/figures#video5

effectively reducing the net apparent sliding rate. Indeed, a simple model that involves two distinct nucleosomal binding sites with different affinities for ALC1, where the second binding event causes an overall decrease in rate, can in principle explain the markedly non-monotonic shape of the titration curves (*Figure 5—figure supplement 1*; see also Materials and methods). However, the lack of numerical stability in fitting this model to our experimental data prevented us from reliably determining the fit parameters. Experimentally testing this hypothesis would require an asymmetric (H3-H4)$_2$ tetramer. In this tetramer, only a single copy of H3 would be fluorescently labeled, and the tetramer would exhibit a defined and known orientation on the DNA, with the labeled H3 proximal to either the long or short linker DNA. However, preparing such a nucleosome is currently not possible (reviewed in *Mitchener and Muir, 2021*). We note that we therefore cannot formally rule out alternative explanations, which do not involve asymmetric nucleosome PARylation, for the observed deviation from Michaelis–Menten kinetics of nucleosome sliding rates. PARylation of the nucleosome did not affect the rate of sliding observed at saturating concentrations of ALC1, but instead enhanced sliding rates by ~2.5 -fold at low concentrations of ALC1 (*Figure 5B*). The overall fastest sliding rate observed in the presence of HPF1 occurs at ~30 nM ALC1, which matches the $K_M$ value determined in the absence of HPF1.

To investigate the role of the H4 tail and acidic patch interactions, we constructed two different FRET-labeled end-positioned nucleosomes in addition to WT: a nucleosome with acidic patch mutations (APM nucleosome; alanine substitutions of H2A E61A, E64A, D90A, and E92A) and a nucleosome with tail-less (globular) H4 histones (gH4 nucleosome; with a deletion of H4 residues 1–19). Additionally, we used ALC1$^{fl}$ with the two substitutions R611A and S612A within the regulatory linker segment (ALC1$^{fl}$ R611A/S612A) to disrupt the acidic patch interaction on the part of ALC1, as done previously (*Lehmann et al., 2020*). Remodeling by ALC1 was consistently slower at all concentrations of ALC1 tested when using ALC1$^{fl}$ R611A/S612A with WT nucleosomes (*Figure 5C*, *Figure 5—figure supplement 2*), ALC1$^{fl}$ with APM nucleosomes (*Figure 5D*, *Figure 5—figure supplement 3*), or ALC1$^{fl}$ with gH4 nucleosomes (*Figure 5E*, *Figure 5—figure supplement 4*; see also *Table 3*).

To determine whether the slower remodeling rates observed with gH4 nucleosomes and ALC1$^{fl}$ R611A/S612A were caused by a defect in coupling ATP hydrolysis to the mechanical translocation of DNA around the histone octamer, we measured the ATPase activity of ALC1 stimulated by PARylated nucleosomes (*Figure 5F*, *Figure 5—figure supplement 5*; see also Materials and methods). Interestingly, the ATPase activity of ALC1 was stimulated to the same extent regardless of whether the PARylation reaction prior to the addition of ATP contained HPF1 or not. This suggests that the faster sliding rate for nucleosomes PARylated with HPF1 observed with low concentrations of ALC1 (*Figure 5B*) was not due to more rapid ATP hydrolysis, but rather can be explained by a higher affinity of ALC1 for the PARylated side of the nucleosome. Both the disruption of the acidic patch interaction by the R611A/S612 mutation of ALC1 and the deletion of the H4 tail caused a decrease in ATPase activity by a factor of ~3.5, suggesting that the acidic patch and the H4 tail both regulate ALC1 remodeling primarily by stimulating its ATPase activity.

## Discussion

Consistent with the involvement of ALC1 in the DDR triggered by PARylation of chromatin, the establishment of a maximally active ALC1-nucleosome complex in vitro required prior PARylation of the nucleosome substrate. Structures of complexes containing post-translationally modified nucleosomes have been determined before

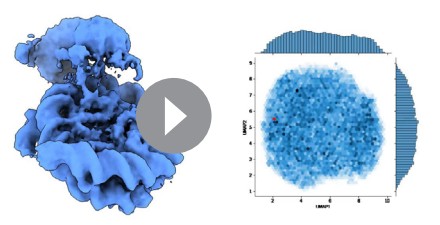

**Video 6.** Dynamics of the H4 tail interaction.
https://elifesciences.org/articles/71420/figures#video6

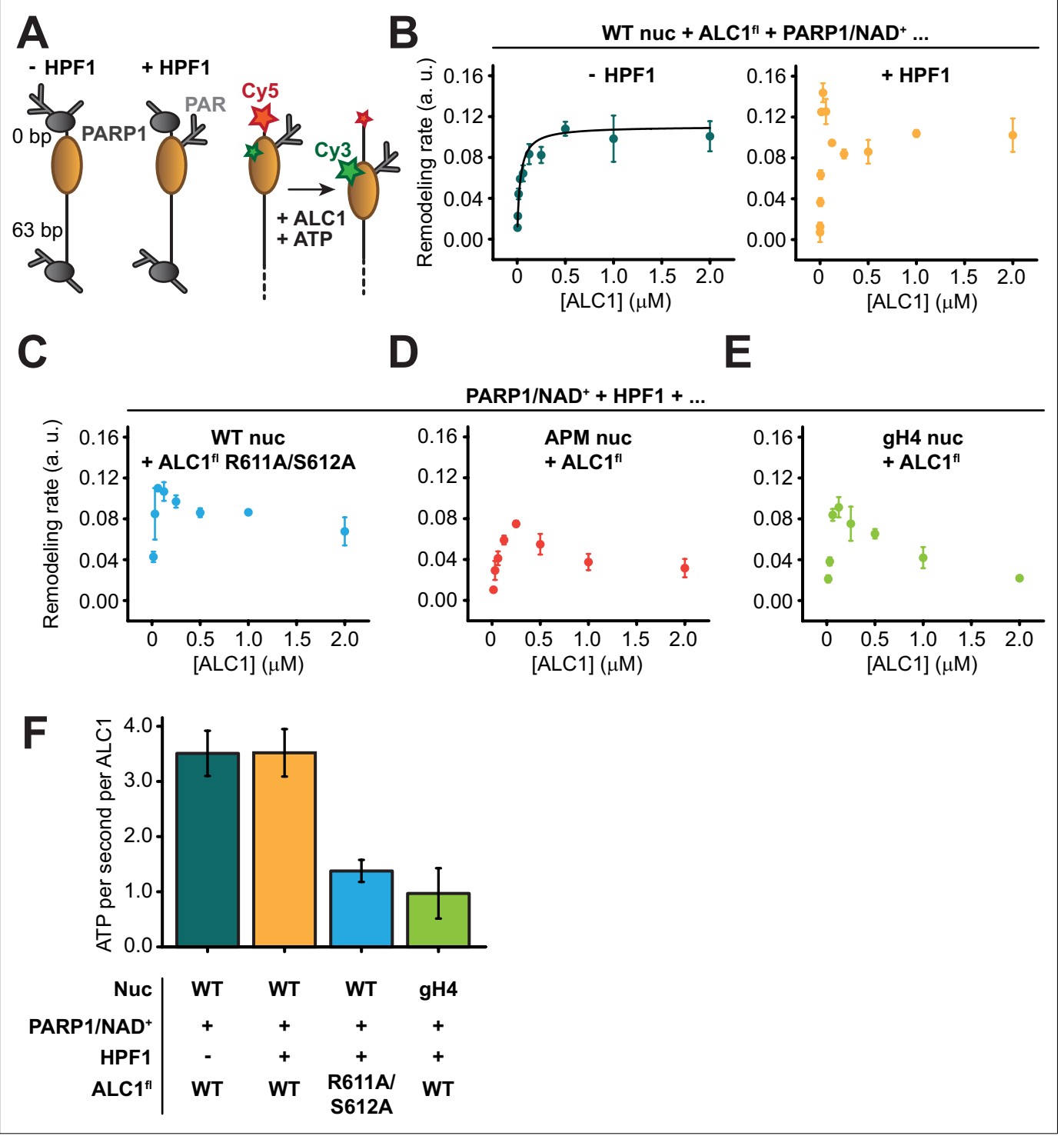

**Figure 5.** ALC1 kinetics in nucleosome sliding and ATPase assays. (**A**) Left: schematic of asymmetric nucleosomes PARylated by PARP1 with and without HPF1. Right: adding ALC1 and ATP to the PARylated FRET-labeled nucleosome results in sliding the histone octamer away from the short linker DNA end, resulting in a decrease of FRET. (**B**) Saturation curves of nucleosome sliding rates for 10 nM WT nucleosomes versus varying concentrations of ALC1fl. Nucleosome sliding assays were performed after PARylation by 80 nM PARP1 with 25 μM NAD+ in the absence (teal) or presence (orange) of 20 nM of HPF1. Points and error bars represent the mean and standard deviation, respectively, from three independent experiments. (**C**) Saturation curve of nucleosome sliding rates for 10 nM WT nucleosomes versus varying concentrations of ALC1fl R611A/S612A. Nucleosome sliding assays were performed after PARylation by 80 nM PARP1 and 20 nM HPF1 with 25 μM NAD+. Points and error bars represent the mean and standard deviation, respectively, from three independent experiments. (**D, E**) Saturation curves of nucleosome sliding rates for 10 nM APM (**D**) and gH4 (**E**) nucleosomes

*Figure 5 continued on next page*

*Figure 5 continued*

versus varying concentrations of ALC1$^{fl}$. Nucleosome sliding assays were performed after PARylation by 80 nM PARP1 and 20 nM HPF1 with 25 μM NAD$^+$. Points and error bars represent the mean and standard deviation, respectively, from three independent experiments. (**F**) ATPase activity for 2 μM ALC1$^{fl}$ (teal, orange, and green) or ALC1$^{fl}$ R611A/S612A (blue) in the presence of 250 nM 10 -N-10 WT nucleosomes (teal, orange, and blue) or 10 -N-10 gH4 nucleosomes (green), activated by PARylation by PARP1 with (orange, blue, and green) or without HPF1 (teal). Bars and error bars represent the mean and standard deviation, respectively, from three independent experiments.

The online version of this article includes the following source data and figure supplement(s) for figure 5:

**Source data 1.** Nucleosome sliding and ATPase data.

**Figure supplement 1.** A simplistic model to reproduce the observed deviation from Michaelis-Menten kinetics.

**Figure supplement 2.** Saturation curve shown in *Figure 5C* (blue) overlaid with the curve for wild-type ALC1 shown in *Figure 5B* (orange).

**Figure supplement 3.** Saturation curve shown in *Figure 5D* (red) overlaid with the curve for wild-type ALC1 shown in *Figure 5B* (orange).

**Figure supplement 4.** Saturation curve shown in *Figure 5E* (green) overlaid with the curve for wild-type ALC1 shown in *Figure 5B* (orange).

**Figure supplement 5.** Continuous kinetic assay to measure ATP consumption.

(*Anderson et al., 2019*; *Chatterjee et al., 2015*; *Hsu et al., 2019*; *Jang et al., 2019*; *Kasinath et al., 2021*; *Lu et al., 2008*; *Morgan et al., 2016*; *Sundaramoorthy et al., 2018*; *Valencia-Sánchez et al., 2021*; *Valencia-Sánchez et al., 2019*; *Wang et al., 2020b*; *Wang et al., 2020a*; *Wilson et al., 2016*; *Worden et al., 2020*, *Worden et al., 2019*; *Xue et al., 2019*; *Yao et al., 2019*), but the present study is, to the best of our knowledge, the first report of a PARylated nucleosome used for structure determination. Cryo-EM data collected on this ALC1-PARylated nucleosome complex in the absence of any cross-linking allowed us to determine the structure of the active state of the ATPase motor of ALC1 bound to the nucleosome. The structure revealed that ALC1 shares two conserved features with other remodelers: its ATPase motor binds to the nucleosomal DNA at SHL2, and the C-ATPase lobe interacts with the N-terminal tail of histone H4. This interaction with the H4 tail was previously shown to be important for remodeling by ALC1, both in vitro using gH4 nucleosomes (*Ahel et al., 2009*) and in cells with the point mutations D377A and D381A in ALC1 (*Verma et al., 2021*; *Figure 2—figure supplement 3*). Because we were able to avoid the use of cross-linking due to stabilization of the ALC1-nucleosome complex by PARylation, our cryo-EM dataset also captured the ensemble of states ALC1 samples during the recognition of a PARylated nucleosome. Analysis of this heterogeneous ensemble with novel computational methods enabled the visualization of these structural states and the possible transitions between them. A subset of the states suggested the location of the macro domain and how it might be positioned to read out PARylation on the H3 tail concomitantly with the ATPase motor engaging its binding site at SHL2. Other states revealed transient interactions between ALC1 and the nucleosome acidic patch, involving a previously identified region of the linker between the ATPase motor and macro domain (*Lehmann et al., 2020*).

Visualization of these nucleosome epitopes interacting with ALC1 prompted us to measure the effect of their perturbation. This indicated that robust interaction of ALC1 with both the H4 tail and the acidic patch is important for nucleosome sliding and that the slower rates of sliding observed upon perturbation of these interactions are caused by slower ATP hydrolysis. Remarkably, the slowest

**Table 3.** Kinetic parameters from nucleosome sliding assays.

Rates are expressed as the mean ± standard deviation from three independent measurements (NA: not applicable).

| Condition | Sliding rate (a. u.) at [ALC1] = 31.25 **nM** | [ALC1] at peak (nM**)** | Sliding rate at peak (a.u**.)** | Sliding rate at saturation ([ALC1] = 2 μM**)** |
|---|---|---|---|---|
| -HPF1 | 0.059 ± 0.002 | NA | NA | 0.101 ± 0.015 |
| + HPF1 | 0.144 ± 0.009 | 31.25 | 0.144 ± 0.009 | 0.102 ± 0.016 |
| + HPF1, ALC1$^{fl}$ R611A, S612A | 0.085 ± 0.025 | 62.50 | 0.110 ± 0.003 | 0.068 ± 0.014 |
| + HPF1, APM nuc | 0.029 ± 0.009 | 250.00 | 0.075 ± 0.003 | 0.032 ± 0.009 |
| + HPF1, gH4 nuc | 0.038 ± 0.004 | 125.00 | 0.091 ± 0.010 | 0.022 ± 0.002 |

rate of ATP hydrolysis measured for ALC1[fl] stimulated by PARylated gH4 nucleosomes is almost twice as fast as the fastest rate of hydrolysis measured previously for the constitutively active ALC1[fl] R860W stimulated by unmodified WT nucleosomes (*Lehmann et al., 2020*). Moreover, the rate of hydrolysis by ALC1[fl] stimulated by PARylated WT nucleosomes is more than sixfold faster than that of ALC1[fl] R860W stimulated by unmodified WT nucleosomes (*Lehmann et al., 2020*). This further emphasizes that the bona fide substrate of ALC1 is a PARylated nucleosome, eliciting an ATP hydrolysis rate on par with that of Chd1 (*Hauk et al., 2010*; *Sundaramoorthy et al., 2017*). Despite these differences in the absolute rate of ATP hydrolysis between these reactions with PARylated or unmodified nucleosomes, the R611A/S612A mutations that disrupt the acidic patch interaction reduce ATP hydrolysis by a similar factor of ~3.5 -fold in both settings.

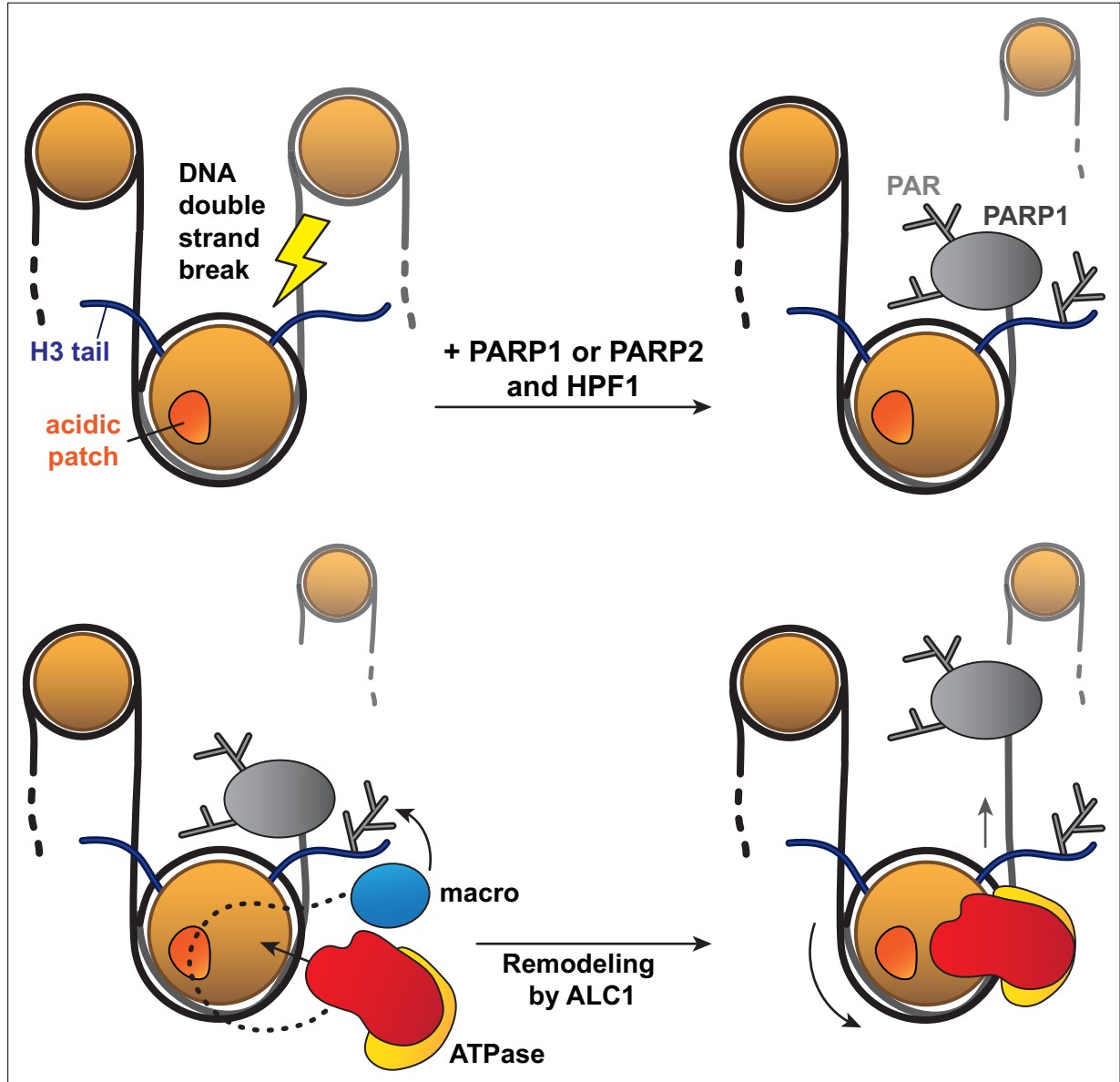

**Figure 6.** Model for the recognition of a PARylated nucleosome by ALC1. Upon DNA damage, PARP1 or PARP2 and HPF1 deposit PAR chains on histones, causing the rapid recruitment of proteins containing PAR reader domains. Among these proteins, ALC1 selectively binds to PAR chains with its macro domain and recognizes the closest nucleosome by probing its acidic patch. Once the macro domain and linker region of ALC1 jointly recognize a PARylated nucleosome, the ATPase motor is released from their auto-inhibitory effect and can tightly bind to the nucleosomal DNA at SHL2 for productive remodeling. Preferential PARylation of target sites spatially closest to the DNA break causes preferential recruitment of ALC1 on the side of the nucleosome such that remodeling slides this nucleosome away from the break. By setting directionality in remodeling, this mechanism could promote exposure of the lesion to downstream repair factors.

While our manuscript was under review, an article was published that reported the crystal structure of ALC1 in its auto-inhibited state and the cryo-EM structure of a constitutively active mutant of ALC1 cross-linked to an unmodified nucleosome (*Wang et al., 2021*). The structure of ALC1 in the nucleosome-free state confirms the model of auto-inhibition that was proposed previously (*Lehmann et al., 2017*; *Singh et al., 2017*) and shows good agreement with one of the states we observed here with ALC1 loosely bound to the nucleosome (*Figure 3—figure supplement 2*). This auto-inhibited structure also reveals important structural details of the interaction between the macro domain and the C-terminal ATPase lobe, which was previously reported as the defining aspect of the auto-inhibited conformation of ALC1 (*Lehmann et al., 2017*). The overall structure of the cross-linked complex between ALC1 and the nucleosome by Wang et al. agrees well with our previous and present findings. In both the structure we report here and the one by Wang et al., ALC1 binds the nucleosome at the same superhelical location and interacts with the H4 tail. The structure by Wang et al. could not resolve the macro domain of ALC1, most likely due to their use of an unmodified nucleosome (*Wang et al., 2021*). In our structure of ALC1 bound to a PARylated nucleosome, we not only detected the macro domain but also its dynamic interaction with the PARylated H3 tail. The structure by Wang et al. also confirms the interaction of the regulatory linker segment of ALC1 with the nucleosome acidic patch that we observed previously (*Lehmann et al., 2020*) as well as in the present study. Interestingly, Wang et al. found two arginine anchors, R611 and R614. In their structure, residue R614 of ALC1 binds to the acidic pocket formed by residues E61, D90, and E92 of histone H2A in a conformation termed 'canonical Arg anchor', while R611 adopts a 'variant Arg anchor' binding mode (reviewed in *McGinty and Tan, 2021*). In our previous structure of the regulatory linker segment of ALC1 cross-linked to a nucleosome, we observed residue R611 as the canonical Arg anchor (*Lehmann et al., 2020*). Such an interaction with the nucleosome acidic patch, involving different Arg anchor residues in different structures, occurred in at least one other instance. The BAH domain of Sir3 contains a stretch of several consecutive arginine residues. In two structures, residue R29 is the canonical Arg anchor (*Armache et al., 2011*; *Arnaudo et al., 2013*), while residue R30 (*Wang et al., 2013*) or R32 (*Yang et al., 2013*) occupies the canonical Arg anchor position in two other structures of the same complex. It is therefore not surprising that a similar observation could be made with residues R611 and R614 of ALC1, especially since they are part of a flexible linker.

Of note, the asymmetric linker DNA length on the 63 N-0 nucleosome construct used for sliding assays appeared to bias PARylation of histones towards the sites closest to the short DNA end, which in turn affected the directionality of nucleosome sliding by ALC1. This in vitro phenomenon may mimic the processing of a DNA break in vivo: in this context, PARP1 or PARP2 would bind to the break and preferentially PARylate target sites closest to it, which would in turn cause ALC1 to slide the PARylated histone octamer away from the DNA break, making it more accessible to downstream repair factors (*Figure 6*).

In conclusion, our structural and biochemical analyses provide a window into the critical early steps by which, in the context of damaged chromatin, ALC1 recognizes and engages PARylated nucleosomes for productive remodeling. Our work sheds light on the intricate regulatory mechanisms that control the activity of the ALC1 remodeler, which is emerging as an important therapeutic option in homologous recombination-deficient cancers.

# Materials and methods

**Key resources table**

| Reagent type (species) or resource | Designation | Source or reference | Identifiers | Additional information |
|---|---|---|---|---|
| Strain, strain background (*E. coli*) | Rosetta 2 (DE3) | Novagen | Cat# 71,400 | Chemically competent |
| Antibody | anti-poly (ADP) ribose polymer (mouse monoclonal) | Abcam | Cat# ab14459, RRID:AB_301239 | WB (1:500) |
| Antibody | Mouse IgG, HRP-linked whole Ab (from sheep) | Cytiva | Cat# NA931, RRID:AB_772210 | WB (1:5000) |

*Continued on next page*

*Continued*

| Reagent type (species) or resource | Designation | Source or reference | Identifiers | Additional information |
|---|---|---|---|---|
| Recombinant DNA reagent | Human ALC1fl (16-879) pNIC-CH2 (plasmid) | *Lehmann et al., 2017* | | UniProt ID Q86WJ1 |
| Recombinant DNA reagent | Human ALC1cat (16-612) pNIC-CH2 (plasmid) | *Lehmann et al., 2017* | | UniProt ID Q86WJ1 |
| Recombinant DNA reagent | Human ALC1macro (613-879) pNIC-CH2 (plasmid) | *Lehmann et al., 2017* | | UniProt ID Q86WJ1 |
| Recombinant DNA reagent | Human HPF1 (1–346) pET28a (plasmid) | *Gaullier et al., 2020* | | UniProt ID Q9NWY4 |
| Recombinant DNA reagent | Human PARP1 (1–1014) pET28a (plasmid) | *Langelier et al., 2017* | | UniProt ID P09874 |
| Recombinant DNA reagent | Human PARP2 Isoform 2 (1-570) pET28a (plasmid) | GenScript | | UniProt ID Q9UGN5 |
| Software, algorithm | RELION-3.1.1 | *Zivanov et al., 2018* | RRID:SCR_016274 | https://www3.mrc-lmb.cam.ac.uk/relion/index.php/Main_Page |
| Software, algorithm | UCSF MotionCor2 version 1.3.2 | *Zheng et al., 2017* | | https://emcore.ucsf.edu/ucsf-software |
| Software, algorithm | CTFFIND4 version 4.1.9 | *Rohou and Grigorieff, 2015* | RRID:SCR_016732 | https://grigoriefflab.umassmed.edu/ctffind4 |
| Software, algorithm | Topaz version 0.2.4 | *Bepler et al., 2019* | | https://github.com/tbepler/topaz, *Tristan, 2021* |
| Software, algorithm | cryoSPARC version 3.2 | *Punjani et al., 2017* | RRID:SCR_016501 | https://cryosparc.com/ |
| Software, algorithm | cryoDRGN version 0.3.2 | *Zhong et al., 2021* | | https://github.com/zhonge/cryodrgn, *Ellen, 2021* |
| Software, algorithm | UCSF ChimeraX 1.1 | *Goddard et al., 2018* | RRID:SCR_015872 | https://www.rbvi.ucsf.edu/chimerax/ |
| Software, algorithm | ISOLDE 1.1 | *Croll, 2018* | | https://isolde.cimr.cam.ac.uk/ |
| Software, algorithm | Phenix version 1.19.2–4158 | *Liebschner et al., 2019* | RRID:SCR_014224 | https://www.phenix-online.org/ |

## Expression and purification of recombinant proteins

All ALC1 constructs were expressed and purified as described previously (*Lehmann et al., 2017*). HPF1 was expressed and purified as described in *Gaullier et al., 2020*. PARP1 and PARP2 were expressed and purified as described in *Langelier et al., 2017*. The pET28a expression vector for PARP2 (isoform two with an N-terminal 6-His tag and thrombin cleavage site) was obtained from GenScript. In brief, proteins were expressed in *E. coli* Rosetta 2 (DE3) by addition of IPTG to the culture media. For PARP1 and PARP2, benzamide was added to the expression cultures. Bacterial pellets were harvested by centrifugation and lysed by sonication, and the lysates were cleared by centrifugation and filtration and loaded on a HisTrap HP 5 ml affinity column (Cytiva). Bound proteins were eluted with imidazole, and fractions of interest were identified by SDS–PAGE and pooled. As a second step, PARP1 and PARP2 were subjected to affinity chromatography again on a HiTrap Heparin HP 5 ml column (Cytiva) and eluted with a gradient to 1 M NaCl. ALC1 constructs were subjected to ion exchange chromatography on a HiTrap Q HP 5 ml anion exchange column (Cytiva) and a HiTrap SP HP 5 ml cation exchange column (Cytiva) mounted in tandem (Q column to trap contaminating DNA, removed from the circuit before eluting the protein from the SP column with a gradient to 1 M NaCl). HPF1 was directly concentrated and subjected to size exclusion chromatography on a Superdex 75 16/60 column (GE Healthcare), without a second affinity or ion exchange step. ALC1 constructs, PARP1, and PARP2 were concentrated and subjected to size exclusion chromatography on a Superdex 200 16/60 column (GE Healthcare). Pure fractions after size exclusion chromatography were identified by SDS–PAGE, pooled, concentrated, and flash frozen in liquid nitrogen before storage at –80 °C.

## Nucleosome preparation

Nucleosomal DNA fragments were prepared following the same general strategy as described before (*Farnung et al., 2017*). In brief, DNA was amplified from a template plasmid containing the 601 sequence (*Lowary and Widom, 1998*) by PCR in 96-well plates using Phusion polymerase, Phusion HF buffer (New England Biolabs) and each primer at 1 µM (Integrated DNA Technologies). For FRET-labeled nucleosomes, the reverse primer (corresponding to the short linker end) was labeled with Cy5. For labeled DNA, the amplified product was purified using a PrepCell (BioRad). For unlabeled DNA, the amplified product was then purified by anion exchange chromatography on a HiTrap Q HP 1 ml column (Cytiva) by loading 10 ml of pooled PCR reaction on the column at 1 ml/min in 50 mM Tri–HCl pH 8, 100 mM NaCl, 0.1 mM EDTA. Elution was performed with a gradient to 1 M NaCl over 20 column volumes. Adequate fractions were identified by native PAGE on a 10 % polyacrylamide gel, pooled, subjected to ethanol precipitation, and dissolved in a small volume of pure water. One PCR plate typically yielded around 0.4 mg of DNA.

Purified *Xenopus laevis* histones were purchased from the Histone Source Protein Expression and Purification Facility, Colorado State University, Fort Collins, CO. For FRET-labeled nucleosomes, histone H2A was labeled with Cy3 at position C120 prior to octamer refolding. The histone octamer was refolded by mixing equimolar amounts of H3, H4, H2A, and H2B dissolved in the unfolding buffer (20 mM Tris–HCl pH 7.5, 6 M guanidine HCl, 5 mM DTT) and dialyzing the mixture against refolding buffer (10 mM Tris–HCl pH 7.5, 2 M NaCl, 1 mM EDTA, 5 mM 2-mercaptoethanol) three times over the course of 20 hr. The resulting histone octamer was concentrated and purified by size exclusion chromatography on a Superdex 200 16/60 column (GE Healthcare), and pure fractions were identified by SDS–PAGE (*Dyer et al., 2004*; *Muthurajan et al., 2016*). Nucleosomes were assembled by mixing equimolar amounts (as determined by small-scale assembly reactions analyzed by native-PAGE) of histone octamer and DNA in high-salt buffer (10 mM Tris–HCl pH 7.5, 2 M NaCl, 1 mM EDTA, 1 mM DTT) and dialyzing continuously to 0 M NaCl (*Dyer et al., 2004*; *Muthurajan et al., 2016*). The nucleosomes were centrifuged at 16,000 × g for 5 min, and the supernatant was transferred to a clean tube, to remove precipitated excess histone octamers. Labeled nucleosomes for FRET experiments were additionally purified using a PrepCell (BioRad).

## Gel shift assay

PARylated 5'-phosphorylated, 5'-biotinylated 10 N-10 nucleosomes were purified from PARP2 and HPF1 by biotin-avidin chromatography prior to gel shift analysis. Unmodified and PARylated nucleosomes were mixed with different ALC1 constructs (ALC1$^{cat}$ residues 16–612; ALC1$^{macro}$ residues 613–879; and ALC1$^{fl}$ residues 16–879) in 1:4 molar ratio in reaction buffer (15 mM HEPES pH 7.5, 50 mM NaCl, 1 mM DTT). The reactions were incubated for 1 hr on ice. Glycerol was added to the final concentration of ~25 % to help load the sample onto 7.5 % Mini-PROTEAN TGX Precast Protein Gel (BioRad). The native PAGE was run in the cold room with 0.25 × TBE running buffer at 100 V for 150 min. The gel was stained with GelGreen and imaged on a BioRad ChemiDoc MP Imaging System and stained further with InstantBlue.

## Ensemble FRET assay for nucleosome remodeling

Nucleosome ensemble remodeling kinetics were measured by monitoring the Cy5 (under 620 nm and 520 nm excitation) and Cy3 (under 520 nm excitation) fluorescence emission signals of a solution of FRET-labeled 63 N-0 nucleosomes using a CLARIOstar (BMG Labtech) multimode microplate reader. Ensemble nucleosome remodeling assays were performed with 10 nM nucleosomes, varying concentrations of ALC1$^{fl}$ as indicated, and 50 µM ATP in remodeling buffer (20 mM HEPES pH 7.5, 50 mM KCl, 6 mM MgCl$_2$, 5 % sucrose, 0.1 mg/ml BSA, 1 mM DTT). Nucleosomes were PARylated prior to nucleosome remodeling by incubating with 80 nM PARP1, 0 or 20 nM HPF1, and 25 µM NAD$^+$ in the remodeling buffer at 37 °C for 5 min. All nucleosome sliding rates were calculated as the slope of a tangent line to the FRET curve at the initial time point. This quantification was chosen because the remodeling curves substantially deviate from a single-exponential model. Such deviation is expected given the complex multi-step nature of the remodeling reaction itself and the non-linearity of the FRET readout.

## Kinetic model

We derived a model similar to the substrate inhibition model developed by Armstrong and Haldane (*Armstrong and Haldane, 1930*; *Reed et al., 2010*). This model assumes that remodeler-nucleosome

binding is at equilibrium and that the Michaelis constants for the two binding sites, $K_1$ and $K_2$, satisfy $K_2 >> K_1$, such that binding can be considered sequential. The model also assumes that [ALC1] >> [nucleosome]. The net apparent sliding rate V is then given by

$$V = \frac{V_2 \times [ALC1]^2 + V_1 \times K_2 \times [ALC1]}{[ALC1]^2 + K_2 \times [ALC1] + K_1 \times K_2}$$

where $V_1$ and $V_2$ are the maximum sliding rates for the singly- and doubly-bound scenarios.

## ATPase assay

ATPase activity was measured using a coupled enzyme system in which regeneration of hydrolyzed ATP is coupled with NADH oxidation as described previously (*Nørby, 1988*). The assay follows NADH absorbance at 340 nm. Final concentrations of 0.45 mM NADH, 1.0 mM phosphoenolpyruvate, 54 U/ml pyruvate kinase (Sigma), and 78 U/ml lactic dehydrogenase (Sigma) were used. For measurements of nucleosome-stimulated ATPase activity, reactions contained 2 µM ALC1$^{fl}$, 250 nM nucleosomes, and 1 mM ATP in a volume of 30 µl in the reaction buffer (20 mM HEPES pH 8, 25 mM NaCl, 75 mM KCl, 1 mM MgCl$_2$). Nucleosomes were PARylated prior measurements with 2 µM PARP1, 0.5 µM HPF1 and 25 µM NAD$^+$ for 5 min at 37 °C. After PARylation, Olaparib was added at a final concentration of 4 µM. Absorbance at 340 nm was monitored at 37 °C using a CLARIOstar microplate reader (BMG Labtech) and clear low-volume 384-well microplates (Greiner). The decrease in the absorbance [$\Delta A_{340}/\Delta t$] is converted to ATPase activity [ATP/s/enzyme] by using 6.22 mM$^{-1}$cm$^{-1}$ as extinction coefficient of NADH at 340 nm.

Importantly, this reporter reaction produces NAD$^+$ (*Figure 5—figure supplement 5*), which in our specific case is an undesired side effect. Since PARylation reactions performed prior to ATP hydrolysis rate measurements contain PARP1, and the extent of PARylation is controlled by providing a limiting amount of NAD$^+$, any newly formed NAD$^+$ from the reporter reaction may be used by PARP1 with two possible, undesired consequences: first, PARP1 may further modify the nucleosome over the course of the reaction, thereby changing the substrate of ALC1 as the reaction of interest progresses; second, in doing so or even simply by hydrolyzing NAD$^+$ to ADP-ribose and nicotinamide, PARP1 may shift the equilibrium of the reporter reaction and artificially increase its rate with no relation to the rate of ATP hydrolysis under study. To control for these effects, we performed the reaction in the absence and presence of Olaparib, a clinical PARP inhibitor. When omitting Olaparib, the observed rate of ATP hydrolysis is indeed faster than in reactions containing the inhibitor added after the intended PARylation has completed, but before addition of ATP (*Figure 5—figure supplement 5*, compare blue curve to yellow and green curves). When omitting NAD$^+$ in the initial PARylation reaction and adding Olaparib before ATP is introduced, ATP hydrolysis occurs at baseline rate during the first 10 min, and eventually speeds up to the same rate as catalyzed by ALC1 (*Figure 5—figure supplement 5*, compare red curve to yellow and green curves). This sudden speed up is likely caused by PARP1 over-coming inhibition once enough NAD$^+$ has accumulated at baseline rate, and ALC1 activating once PARP1 has produced PAR chains long enough, which would presumably happen fast since PAR chains only three ADP-ribose units long are sufficient for full activation of ALC1 (*Singh et al., 2017*).

## Preparation of PARylated nucleosomes

Initial PARylation conditions were chosen to form a fully saturated HPF1-PARP2-nucleosome or HPF1-PARP1-nucleosome complex, based on the $K_D$ values for the PARP2-nucleosome and HPF1-PARP2-nucleosome interactions reported in *Gaullier et al., 2020*, and based on reaction conditions reported in *Bilokapic et al., 2020*. We used a nucleosome with 10 bp linker DNA ends on both sides (10 N-10). For reactions with PARP2, we used the same nucleosome with a terminal 5'-phosphate group at one end (5'P-10-N-10), since PARP2 requires it for binding and activation (*Langelier et al., 2014*; *Obaji et al., 2018*). Reactions were initiated by the addition of NAD$^+$ after pre-formation of the HPF1-PARP-nucleosome complex, and reaction products were analyzed by Western blot using a mouse mono-clonal Anti-Poly (ADP-Ribose) Polymer (Abcam, ab14459) antibody. Titrations of NAD$^+$ and PARP/HPF1 were analyzed to determine the concentrations of components giving a strong PARylation signal on histones while keeping auto-PARylation of PARP as limited as possible. Nucleosomes PARylated with PARP2 displayed shorter PAR chains, as assessed from screening micrographs, and were therefore

most suitable for cryo-EM, while nucleosomes for remodeling assays and ATPase assays were PARylated with PARP1 because they did not have a terminal 5'-phosphate group required by PARP2.

## Western blot

Reactions were prepared in 30 µl in the assay buffer (25 mM HEPES-NaOH pH 8.0, 50 mM NaCl, 0.1 mM EDTA, 0.1 mM TCEP). Nucleosomes at 1 µM final concentration were incubated with PARP1 or PARP2 and HPF1 in the molar ratios as indicated above the panels (*Figure 1—figure supplement 3*). For the initial three experiments; varying HPF1, PARP1 and PARP1:HPF1 amounts (panels 1–3), NAD$^+$ was added at the final concentration of 6.25 mM. For NAD$^+$ titration, NAD$^+$ was added in the gradient from 0.1 to 12.8 mM (panel 4) and from 150 to 225 µM in 25 µM steps (panel 5). Reactions were incubated 30 min on ice or 10 °C as indicated. After incubation time, reactions were split and loaded onto two identical 4–20% Mini-PROTEAN TGX Precast Protein Gels (Bio-Rad). Proteins were separated by SDS–PAGE at 120 V for 90 min. One gel was stained with InstantBlue Coomassie stain and the other was used for protein immunoblotting. Proteins were transferred onto 0.2 µm nitrocellulose blotting membranes (Cytiva) by wet electro transfer at constant 20 V overnight in the cold room. Transfer efficiency was confirmed by staining with 0.1% w/v Ponceau in 5 % acetic acid. Membranes were blocked with 5 % skim milk/TBS-T (1× TBS Tween-20) for 1 hr at room temperature and probed with mouse monoclonal Anti-Poly (ADP-Ribose) Polymer (Abcam, ab14459) primary antibody overnight at 4 °C. Membranes were then washed three times for 5 min with TBS-T, incubated with sheep anti-mouse (Cytiva, NA931) secondary antibody conjugated to a horseradish peroxidase (HRP) for 1 hr at room temperature, and washed again three times for 5 min with TBS-T. Immunoblots were developed using Clarity Western ECL Substrate (Bio-Rad).

## Cryo-EM sample preparation

The sample was prepared as follows: an initial mixture of 5'P-10-N-0 nucleosome at 1 µM, PARP2 at 1 µM, and HPF1 at 4 µM was incubated on ice for 30 min before addition of NAD$^+$ to 200 µM and further incubation at 10 °C for 60 min to allow PARylation until the reaction ends (either because it exhausts the limiting amount of NAD$^+$ or because auto-modified PARP2 dissociates from the nucleosome). ALC1$^{fl}$ was then added to a final concentration of 3 µM, and ADP-BeF$_3$ to a final concentration of 1 mM (1 ×). The mixture was further incubated on ice for 50 min before vitrification. ADP-BeF$_3$ was prepared as a 10 × stock (10 mM ADP, 30 mM BeSO$_4$, 150 mM NaF, 10 mM MgCl$_2$), freshly before use.

Quantifoil R 2/2 Cu 200 grids (Electron microscopy sciences) were glow-discharged at 20 mA and 0.4 mbar with negative polarity for 60 s in a PELCO easiGlow glow discharger. Three microliters of the sample was applied onto grids and immediately blotted for 2.5 s. Grids were plunge-frozen into liquid ethane using a Vitrobot Mark IV (Thermo Fisher Scientific) operated at 100 % relative humidity and 4 °C.

## Cryo-EM data collection and processing

Cryo-EM data were collected at the SciLifeLab facility in Stockholm, Sweden, on a Titan Krios equipped with a Gatan K3 detector operated in counting mode and an energy filter with a slit width of 20 eV. Magnification was 105 kx, resulting in an image pixel size of 0.84 Å/pixel. A total accumulated dose of 45 e$^-$/Å$^2$ was fractionated in 40 movie frames. Movies were motion-corrected using UCSF MotionCor2 version 1.3.2 (*Zheng et al., 2017*), and CTF parameters were estimated using CTFFIND4 version 4.1.9 (*Rohou and Grigorieff, 2015*), both from within RELION version 3.1.1 (*Zivanov et al., 2018*). Particle-picking was done using Topaz version 0.2.4 (*Bepler et al., 2019*). A relatively homogeneous set of 43,698 particles could be isolated by several rounds of 2D and 3D classification in RELION. The 3D reference used for the first 3D classification was a synthetic map generated from a nucleosome atomic model from PDB entry 3LZ0 (*Vasudevan et al., 2010*) using a low-pass filter to 30 Å resolution. Subsequent 3D classifications used the best map from the previous round as 3D reference. Details are provided in *Figure 2—figure supplement 1*.

## Identification and 3D refinement of the active state of the ALC1-nucleosome complex

The map resulting from 3D refinement of the 43,698 particles showed signs of continuous conformational heterogeneity in the region of ALC1, notably with alpha helices appearing as flattened stretches

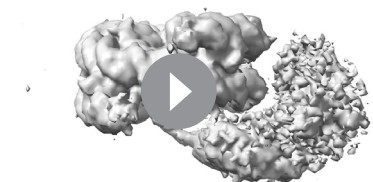

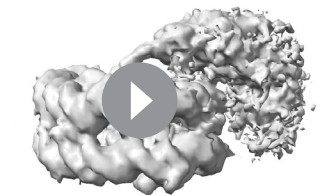

**Video 7.** Traversal of the first principal component of variability in the Z values assigned by cryoSPARC 3DVA, with maps shown in dyad view.
https://elifesciences.org/articles/71420/figures#video7

**Video 9.** Traversal of the first principal component of variability in the Z values assigned by cryoSPARC 3DVA, with maps shown in gyres view.
https://elifesciences.org/articles/71420/figures#video9

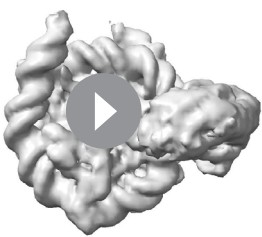

**Video 8.** Traversal of the first principal component of variability in the Z values assigned by cryoSPARC 3DVA, with maps shown in disc view.
https://elifesciences.org/articles/71420/figures#video8

of density instead of cylinders (as alpha helices in histones). Accordingly, attempts at improving the ALC1 density using multi-body refinement in RELION (*Nakane et al., 2018*) were unsuccessful, yielding only deteriorated maps for this body. This further suggested that this dataset does not meet the assumption underlying this approach that the whole particle is composed of several rigid bodies, meaning that ALC1 undergoes conformational changes. We imported these particles into cryoSPARC version 3.2 (*Punjani et al., 2017*) at their original pixel size of 0.84 Å/pixel with a box size of 512 pixels, and Fourier cropped them to a box size of 400 pixels. We ran homogeneous refinement with automatic masking to generate an appropriate mask for 3D variability analysis (3DVA). We then subjected these particles to 3DVA (*Punjani and Fleet, 2021*), solving for three principal components and with a filter resolution of 4 Å; all other parameters were left with their default values. The resulting distributions along the principal components of variability were smooth, confirming both the compositional homogeneity of this set of particles and the presence of conformational flexibility (*Figure 2—figure supplement 1*). Notably, the trajectory along PC1 revealed that the first source of variability in the dataset is the presence of two binding modes of ALC1 to the nucleosomes: one in which the ATPase motor is loosely bound to the DNA and the remodeler makes an interaction with the acidic patch, the other in which the ATPase motor is tightly associated with the DNA at SHL2 and with no visible interaction with the acidic patch (*Videos 7–9*). We call this latter state the 'active state', based on previous literature on chromatin remodelers and since this tightly bound ATPase motor is a requirement for remodeling activity. Clustering particles based on their latent coordinates from 3DVA into five subpopulations isolated a set of 5487 particles that gave an improved reconstruction of the active state of ALC1. These particles were finally subjected to non-uniform refinement (*Punjani et al., 2020*), which yielded a map of the active state with a global resolution of 4.8 Å (*Figure 2A*) and small variations in local resolution (*Figure 2—figure supplement 1*).

## Analysis of heterogeneity in the cryo-EM dataset

Given the evidence that the set of 43,698 still contained significant continuous heterogeneity, we then analyzed it with cryoDRGN version 0.3.2 (*Zhong et al., 2021*). All models were trained for 50 epochs

on a single GPU. We re-extracted these particles in RELION with a box size of 128 pixels and a pixel size of 3.36 Å/pixel for rapid evaluation. We first trained a cryoDRGN model on these downsampled particles with three layers and 256 nodes per layer for both the encoder and decoder network architectures (termed 256 × 3) and a one-dimensional latent variable Z. This resulted in three major populations of particles, based on Z values (*Figure 3—figure supplement 3*). Evaluating the decoder at the most populated Z values yielded recognizable maps of the complex. These maps were very similar to the average map from consensus refinement (*Figure 3A*), indicating that none of these populations were contaminants or damaged particles and confirming that our strict selections between rounds of 3D classifications yielded a clean set of particles. Next, we trained a 256 × 3 model on these same particles, this time with an eight-dimensional latent variable Z to analyze heterogeneity. We trained this model three times with independent initializations. For all three replicate runs, the resulting distribution of Z values is smooth, indicating that heterogeneity in the dataset originates purely from continuous conformational changes and not from populations of particles with distinct compositions, in agreement with the 3DVA results. We then sampled the resulting distributions of Z values using k-means clustering with 20 cluster centers, as implemented in cryoDRGN. The resulting maps show a variety of states, including states with a tightly bound ATPase domain similar to the active state we could build an atomic model for, and states with a loosely bound ATPase motor and visible interaction with the acidic patch. Finally, we re-extracted the particles with a box size of 128 pixels and a pixel size of 1.68 Å/pixel to get higher resolution reconstructions. This resulted in a smaller physical box size (128 pixels at 1.68 Å/pixel, resulting in a Box 215 Å wide), which in this case was beneficial since the initial box (128 pixels at 3.36 Å/pixel, resulting in a Box 430 Å wide) was wide enough to enclose neighboring particles. A trial-and-error exploration of box size and pixel size parameters gave more interpretable maps when using the same box size in pixels with smaller pixels (resulting in a smaller physical box size) than when compensating for smaller pixels by increasing the box size in pixels in order to keep the same physical box size. Tighter boxes tended to give better results because the encoder network of cryoDRGN is presented with the entire, unmasked box, and larger boxes were more likely to contain neighboring particles around the particle of interest at the center of the box, artificially increasing the heterogeneity detected by the encoder.

## Model building and refinement

We generated a homology model of the ALC1 ATPase domain using SWISS-MODEL (*Waterhouse et al., 2018*). The structure of ISWI from PDB entry 6JYL (*Yan et al., 2019*) provided the template that gave the best-scoring model according to SWISS-MODEL's metrics. We used the nucleosome model from PDB entry 6RYR (*Farnung et al., 2020*). These two models were placed into the map of the active state using UCSF ChimeraX version 1.1 (*Goddard et al., 2018*; *Pettersen et al., 2021*) and subjected to three rounds of interactive molecular dynamics flexible fitting (iMDFF) using ISOLDE version 1.1 *Croll, 2018* followed by real-space refinement in phenix.real_space_refine (*Afonine et al., 2018*). Model-to-map real-space correlation coefficients and model geometry statistics were calculated with phenix.validation_cryoem. All Phenix programs used were from the Phenix suite version 1.19.2–4158 (*Liebschner et al., 2019*).

## Acknowledgements

Cryo-EM data were collected at the Swedish National Cryo-EM Facility funded by the Knut and Alice Wallenberg, Family Erling Persson and Kempe Foundations, SciLifeLab, Stockholm University, and Umeå University. We thank M Carroni, K Wallden, and J Conrad for assistance with cryo-EM data collection, and H Schüler for discussions and for providing reagents. We thank J Pascal for generously providing the PARP1 expression vector. The HPF1 expression vector was a kind gift from K Luger. We thank M Schattenhofer for assistance with figure layout. This work was supported by the European Research Council (ERC Starting Grant 714068), the Knut and Alice Wallenberg Foundation (Grant KAW 019.0306), the Swedish Research Council (VR Grant 2019–03534), and Cancerfonden (Grant 19 0055 Pj). SD is an EMBO Young Investigator. GH and SJB are supported by the Francis Crick Institute, which receives its core funding from Cancer Research UK (FC0010048), the UK Medical Research Council (FC0010048), and the Wellcome Trust (FC0010048); SJB is also funded by a European Research Council (ERC) Advanced Investigator Grant (TelMetab) and Wellcome Trust Senior Investigator and Collaborative Grants.

# Additional information

## Competing interests

Simon J Boulton: is co-founder and VP Science Strategy at Artios Pharma Ltd. Sebastian Deindl: Reviewing editor, *eLife*. The other authors declare that no competing interests exist.

## Funding

| Funder | Grant reference number | Author |
|---|---|---|
| European Research Council | 714068 | Sebastian Deindl |
| Knut och Alice Wallenbergs Stiftelse | KAW 019.0306 | Sebastian Deindl |
| Vetenskapsrådet | VR Grant 2019-03534 | Sebastian Deindl |
| Cancerfonden | 19 0055 Pj | Sebastian Deindl |
| Francis Crick Institute | | Graeme Hewitt Simon J Boulton |
| European Research Council | Advanced Investigator Grant | Simon J Boulton |
| Wellcome Trust | | Simon J Boulton |
| Cancer Research UK | FC0010048 | Graeme Hewitt Simon J Boulton |

The funders had no role in study design, data collection and interpretation, or the decision to submit the work for publication.

## Author contributions

Luka Bacic, Data curation, Formal analysis, Investigation, Resources, Validation, Visualization, Writing – review and editing; Guillaume Gaullier, Conceptualization, Data curation, Formal analysis, Investigation, Methodology, Resources, Supervision, Validation, Visualization, Writing - original draft, Writing – review and editing; Anton Sabantsev, Formal analysis, Methodology, Supervision, Validation, Writing – review and editing; Laura C Lehmann, Data curation, Resources, Validation, Visualization, Writing – review and editing; Klaus Brackmann, Resources; Despoina Dimakou, Investigation; Mario Halic, Supervision, Writing – review and editing; Graeme Hewitt, Resources, Writing – review and editing; Simon J Boulton, Funding acquisition, Resources, Writing – review and editing; Sebastian Deindl, Conceptualization, Funding acquisition, Project administration, Resources, Supervision, Validation, Writing – review and editing

## Author ORCIDs

Luka Bacic ![ORCID] http://orcid.org/0000-0001-6896-3506
Guillaume Gaullier ![ORCID] http://orcid.org/0000-0003-3405-6021
Anton Sabantsev ![ORCID] http://orcid.org/0000-0002-8559-8894
Laura C Lehmann ![ORCID] http://orcid.org/0000-0003-2518-5606
Despoina Dimakou ![ORCID] http://orcid.org/0000-0002-1424-5469
Mario Halic ![ORCID] http://orcid.org/0000-0002-0061-7372
Sebastian Deindl ![ORCID] http://orcid.org/0000-0001-6807-8654

## Decision letter and Author response

Decision letter https://doi.org/10.7554/eLife.71420.sa1
Author response https://doi.org/10.7554/eLife.71420.sa2

# Additional files

## Supplementary files
• Transparent reporting form

## Data availability

The cryo-EM map of the ALC1-nucleosome complex in the active state was deposited at the EMDB with accession code EMD-13065. The model of the ALC1-nucleosome complex in the active state was deposited at the PDB with accession code 7OTQ. The map series from the cryoDRGN graph traversal was deposited at the EMDB with accession code EMD-13070. Raw movies, extracted particles and their coordinates, and cryoDRGN and cryoSPARC job directories were deposited in EMPIAR with accession code EMPIAR-10739.

The following datasets were generated:

| Author(s) | Year | Dataset title | Dataset URL | Database and Identifier |
|---|---|---|---|---|
| Bacic L, Gaullier G, Deindl S | 2021 | Cryo-EM structure of ALC1/CHD1L bound to a PARylated nucleosome | https://www.emdataresource.org/EMD-13065 | EMDataResource, EMD-13065 |
| Bacic L, Gaullier G, Deindl S | 2021 | Cryo-EM structure of ALC1/CHD1L bound to a PARylated nucleosome | https://www.rcsb.org/structure/7OTQ | RCSB Protein Data Bank, 7OTQ |
| Bacic L, Gaullier G, Deindl S | 2021 | cryoDRGN graph traversal of the ALC1 - PARylated nucleosome complex particle distribution | https://www.emdataresource.org/EMD-13070 | EMDataResource, EMD-13070 |
| Bacic L, Gaullier G, Deindl S | 2021 | Single-particle cryo-EM dataset of the complex between ALC1 and a PARylated nucleosome | https://www.ebi.ac.uk/empiar/EMPIAR-10739/ | Electron Microscopy Public Image Archive, EMPIAR-10739 |

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
