## [Decision Letter]

**Acceptance summary:**

This paper is of interest to chromatin remodeling and DNA damage repair fields. The native structure of ALC1 bound to a PARylated nucleosome highlights an ensemble of heterogeneous conformations sampled by ALC1 as it binds to DNA and histone surfaces of the nucleosome. Most of the key claims of the paper are well supported by the data.

**Decision letter after peer review:**

Thank you for submitting your article "Structure and dynamics of the chromatin remodeler ALC1 bound to a PARylated nucleosome" for consideration by *eLife*. Your article has been reviewed by 3 peer reviewers, one of whom is a member of our Board of Reviewing Editors, and the evaluation has been overseen by Cynthia Wolberger as the Senior Editor. The reviewers have opted to remain anonymous.

Essential revisions:

1. Can the authors show the PARylation status of histones and PARP1 in their experimental conditions for the sliding assay performed with and without HPF1? It seems that the authors rely on the literature to say that histones should be modified in the presence of HPF1, but it would be important to show it.

Relates to this section:

"We tested the effect of including or omitting HPF1 in this reaction, leading to PAR chains on both histones and PARP1 or on PARP1 alone, respectively (Bonfiglio et al., 2017). In each case, we first allowed the PARylation reaction to deplete a limiting amount of NAD+ (Figure S1A), and then added ALC1 and ATP to initiate sliding. Nucleosome sliding by ALC1 was faster (2- to 4-fold) when PAR chains were attached to histones, rather than exclusively to PARP1 (Figure 1B), indicating the PARylated nucleosome is the proper substrate of ALC1."

2. Can the authors explain better, perhaps in the legend, their rationale for Figure S1B? The authors say that they "systematically assess PARylation conditions to determine the concentrations of PARP2, HPF1 and NAD+ that maximize PARylation of histones but also limit the auto-modification of PARP2". Figure S1B shows systematic variations of reactions involving PARP1, except for a single experiment with PARP2 where NAD+ is titrated. The wording should be changed to reflect the actual screening done. Most of the optimization has been done on PARP1, yet, the cryo-EM complex is assembled with PARP2. Also, with PARP2, the only optimization that has been done is modifying the concentration of NAD+, which does not look to have a large impact on the amount of histones PARylated as shown in their Western Blot. It is also confusing why the two temperatures were chosen.

3. Model bias is a well-known pitfall in cryo-EM. The authors used a synthetic initial 3D reference derived from an atomic model and not from the data themselves, presumably due to limitations in overcoming incorrect local minima. Still, it is now commonplace to generate an initial 3D model from particles or 2D averages. The authors should discuss their attempts at 3D classification using alternatively derived initial references.

4. The results obtained in the following study should be discussed and put into context with the current work:

Nature Communications, Structural basis of ALC1/CHD1L autoinhibition and the mechanism of activation by the nucleosome.

5. In the last paragraph of the introduction, the authors claim "to directly visualize multiple conformational states visited during the early events of substrate recognition". Given the challenges with assigning temporal or kinetic information from cryoDRGN analysis as mentioned elsewhere in the manuscript, I would advise editing this sentence to remove temporal assumptions.

6. In describing Figure 1B, the authors claim 2- to 4- fold increase in nucleosome sliding but it is unclear how this range was selected without any fitting of the curves. Please provide additional information.

7. On page 10, the authors claim that PARylation of histones appears to decrease the Km of ALC1 relative to PARylation of PARP1 alone. I am not convinced that this claim can be made without fitting the forward and reverse remodeling rates successfully and this statement either requires further justification or removal. Potentially this could be justified using theoretical fits as in Figure S5A.

---

## [Author Response]

Essential revisions:1. Can the authors show the PARylation status of histones and PARP1 in their experimental conditions for the sliding assay performed with and without HPF1? It seems that the authors rely on the literature to say that histones should be modified in the presence of HPF1, but it would be important to show it.Relates to this section:"We tested the effect of including or omitting HPF1 in this reaction, leading to PAR chains on both histones and PARP1 or on PARP1 alone, respectively (Bonfiglio et al., 2017). In each case, we first allowed the PARylation reaction to deplete a limiting amount of NAD+ (Figure S1A), and then added ALC1 and ATP to initiate sliding. Nucleosome sliding by ALC1 was faster (2- to 4-fold) when PAR chains were attached to histones, rather than exclusively to PARP1 (Figure 1B), indicating the PARylated nucleosome is the proper substrate of ALC1."

We thank the reviewers for this important comment. Indeed, we initially relied on the literature to assume that the reactions containing HPF1 resulted in the PARylation of histones and of PARP1, while reactions devoid of HPF1 resulted in the PARylation of PARP1 only but not of histones. We believe this was very convincingly demonstrated by a study already cited in the manuscript (Bonfiglio et al., 2017, https://doi.org/10.1016/j.molcel.2017.01.003). Based on the reviewers’ comment, we have now performed a control experiment to verify this in our experimental setup. We replicated the nucleosome sliding experiment shown in Figure 1B but scaled up the PARylation reactions performed prior to the nucleosome sliding measurements. This allowed us to use a fraction of the reaction products to measure nucleosome sliding by ALC1, and to subject the rest of the PARylation reaction products to Western blotting using an α-PAR antibody. This strategy enables two readouts for the same PARylation reaction products: (1) their ability to stimulate nucleosome sliding by ALC1 and (2) the detection of PAR chains present on the reaction components. These analyses confirm that the HPF1-containing reaction not only displayed more rapid ALC1-induced nucleosome sliding but also resulted in histone PARylation. In contrast, such histone PARylation could not be detected in the absence of HPF1, where nucleosome sliding by ALC1 occurred at reduced rates. We have now included these new data as Figure 1—figure supplement 1 and described them in the revised version of the manuscript (page 4, ultimate paragraph).

2. Can the authors explain better, perhaps in the legend, their rationale for Figure S1B? The authors say that they "systematically assess PARylation conditions to determine the concentrations of PARP2, HPF1 and NAD+ that maximize PARylation of histones but also limit the auto-modification of PARP2". Figure S1B shows systematic variations of reactions involving PARP1, except for a single experiment with PARP2 where NAD+ is titrated. The wording should be changed to reflect the actual screening done. Most of the optimization has been done on PARP1, yet, the cryo-EM complex is assembled with PARP2. Also, with PARP2, the only optimization that has been done is modifying the concentration of NAD+, which does not look to have a large impact on the amount of histones PARylated as shown in their Western Blot. It is also confusing why the two temperatures were chosen.

We completely agree with the reviewers that our original description of these results is rather confusing. To clarify, the systematic part of this optimization procedure was carried out with PARP1 at a time when we were still establishing the purification of PARP2. Once we eventually purified PARP2, we chose to directly transfer one of the conditions identified with PARP1, and to vary experimental parameters for PARP2 within a narrower range only. Our trials with PARP1 showed that the ratio between PARP1 and HPF1 strongly influences the distribution of PAR chain lengths, a phenomenon that has been reported by others (Langelier et al., 2021, https://doi.org/10.1101/2021.05.19.444852; Mohapatra et al., 2021, https://doi.org/10.1101/2021.07.06.449314). We chose to test one PARP2:HPF1 ratio only, instead of performing independent titrations of PARP2 and HPF1; we chose an excess of HPF1 to restrict chain length. The next parameter that most influenced the distribution of PAR chain lengths was the concentration of NAD^+^, which we titrated across a narrower range in the experiments with PARP2. The two incubation temperatures were tested as an additional attempt to control PAR chain length, and 10ºC seemed to slightly restrict chain elongation compared to 4ºC. All our attempts were aimed at limiting the chain length since as few as 3 ADP-ribose units are sufficient for tight binding and full activation of ALC1 (Singh et al., 2017, https://doi.org/10.1016/j.molcel.2017.11.019). Moreover, a small cryo-EM dataset collected with samples prepared by PARylating nucleosomes with PARP1 under conditions that generated long PAR chains proved extremely difficult to analyze due to their heterogeneity. We have now revised the manuscript to better explain our *rationale* (page 5, ultimate paragraph).

3. Model bias is a well-known pitfall in cryo-EM. The authors used a synthetic initial 3D reference derived from an atomic model and not from the data themselves, presumably due to limitations in overcoming incorrect local minima. Still, it is now commonplace to generate an initial 3D model from particles or 2D averages. The authors should discuss their attempts at 3D classification using alternatively derived initial references.

We completely agree with the reviewers on the potential risk of model bias in cryoEM.

We did not initially try to generate an *ab initio* model from the data but instead used a synthetic reference as described in the manuscript. We preferred the use of a synthetic reference because we suspected that the high degree of conformational heterogeneity visible in the 2D class averages would yield very poor initial models. As a control, we have now generated four initial models from a random subset of 50 000 particles taken from the set of particles retained after 2D classification and before any 3D classification (Author response image 1) . These models are shown below in two orientations roughly equivalent to the disc view and the gyres view in Figure 2 (the dyad view is not shown because the smeared density visible at the top of the disc view renders the dyad view completely uninformative).

**Author response image 1. sa2fig1:** Four *ab initio* models generated from a random subset of 50 000 particles taken from the particles retained after 2D classification. Top row shows models in disc view, bottom row shows the same models in gyres view.

All these models show a distinctive nucleosome-like shape and smeared additional density near the nucleosome dyad, and are expectedly highly consistent with the 2D class averages. We believe that priming the 3D classification with such models would most likely not allow us to identify a subset of particles similar to the one we eventually used and reported in the manuscript. Most importantly, we believe that our cryoEM map is free of model bias for the following reasons:

1. 2D classification in RELION is always performed without a reference, therefore without any potential for model bias. The 2D class averages from our dataset show very clearly a nucleosome and additional density on only one side of the nucleosome, therefore attributable to a binding factor (Figure 2—figure supplement 1). All these features could arise from the data only, since no reference model was used in this approach. Since the reference-free 2D classification established, without any possible model bias, that the micrographs contain nucleosomes, it seemed reasonable to subsequently use a synthetic map generated from a free nucleosome structure to prime 3D classification.

2. The synthetic map we used as a 3D reference for 3D classification was generated from an atomic model of a free nucleosome (PDB entry 3LZ0). Such a 3D reference, devoid of density extending from the nucleosome, could not have caused spurious alignment of background noise such that it would give rise to density resembling that of a binding factor. In addition, this synthetic map was low-pass filtered to 30 Å. Low-pass filtering of the 3D reference to a sufficiently low resolution is now common practice in cryo-EM, for any type of 3D reference (synthetic or generated ab initio from the data). In our specific case, a low-pass filter to 30 Å only retained the global disk shape of the nucleosome and the handedness of the DNA superhelix around the histone octamer, but did not preserve any secondary structure features (Author response image 2). With such a minimal, low-resolution reference to prime 3D classification, the appearance of additional density corresponding to a binding factor with clear secondary structure features is extremely unlikely to arise from model bias.

**Author response image 2. sa2fig2:** Synthetic 3D reference used to prime 3D classification. This reference was generated from an atomic model of a free nucleosome (PDB entry 3LZ0) with a low-pass filter to 30 Å resolution. The model is shown in dyad view (left), disc view (center) and gyres view (right).

To further verify the absence of model bias in our reconstruction, we performed a control analysis as follows: we removed the Euler angle assignments from the 43 698 particles that yielded the map shown in Figure 3A, only retaining their in-plane rotation angle from 2D classification. This effectively reverted this set of particles to their state after 2D classification. We then calculated an *ab initio* model (Author response image 3) from these unaligned particles. This model shows recognizable nucleosome and binding factor density.

**Author response image 3. sa2fig3:** *Ab initio* model generated from the 43 698 particles after removal of their Euler angles. The model is shown in dyad view (left), disc view (center) and gyres view (right).

Finally, we subjected this set of particles to 3D refinement, using the above *ab initio* model as the initial reference (Author response image 3) . The refinement converged to a reconstruction identical to the one presented in Figure 3A in the manuscript (Author response image 4) .

**Author response image 4. sa2fig4:** Map from 3D refinement of the 43 698 particles primed with the *ab initio* model in Author response image 3. This map is shown at the same contour level as the map in Figure 3A.

We believe that this control and the above considerations rule out the possibility of model bias in this set of 43 698 particles, and consequently in our structural results (the reconstruction in Figure 2 and the heterogeneity analysis in Figure 3 and Figure 4), since they were all derived from the same set of particles or a subset thereof.

4. The results obtained in the following study should be discussed and put into context with the current work:Nature Communications, Structural basis of ALC1/CHD1L autoinhibition and the mechanism of activation by the nucleosome.

We thank the reviewer for pointing out this recent article, which was published after the submission of our manuscript. We have now cited and discussed this study in our manuscript (page 12, ultimate paragraph).

5. In the last paragraph of the introduction, the authors claim "to directly visualize multiple conformational states visited during the early events of substrate recognition". Given the challenges with assigning temporal or kinetic information from cryoDRGN analysis as mentioned elsewhere in the manuscript, I would advise editing this sentence to remove temporal assumptions.

In the revised version of the manuscript, we have now edited this statement as suggested by the reviewers (page 3, ultimate paragraph).

6. In describing Figure 1B, the authors claim 2- to 4- fold increase in nucleosome sliding but it is unclear how this range was selected without any fitting of the curves. Please provide additional information.

Due to the complex multi-step nature of the remodeling reaction itself (large number of sliding intermediates and their reversible interconversion) and the non-linearity of the FRET readout, the observed remodeling curves substantially deviate from a single-exponential decay model. We therefore calculated all nucleosome sliding rates as the slope of a tangent line to the FRET curve at the initial time point, which we believe yields better estimates of the initial sliding rates in comparison to fitting a single-exponential decay curve.

With this definition of the nucleosome sliding rate, we obtain a 2.4-fold faster rate for the “+HPF1” condition compared to the “-HPF1” condition presented in Figure 1B. We have now revised the manuscript to correctly reflect this (see page 5, first paragraph).

7. On page 10, the authors claim that PARylation of histones appears to decrease the Km of ALC1 relative to PARylation of PARP1 alone. I am not convinced that this claim can be made without fitting the forward and reverse remodeling rates successfully and this statement either requires further justification or removal. Potentially this could be justified using theoretical fits as in Figure S5A.

Our initial claim was based on the fact that the remodeling rates at low ALC1 concentrations are consistently higher for PARylated nucleosomes when compared to unmodified nucleosomes, whereas the remodeling rates at saturating ALC1 concentrations are similar. However, we agree with the reviewers that this effect cannot be unambiguously assigned to a difference in K_M_ without successfully fitting the curve given by the remodeling rates as a function of ALC1 concentration. As we stated in the original version of the manuscript, we were not able to obtain reliable fits with the theoretical model in Figure 5—figure supplement 1, likely because the reaction is substantially more complex than this minimalistic model. We have therefore removed the statement in question (see page 10, first paragraph).